# SuperDendrix algorithm integrates genetic dependencies and genomic alterations across pathways and cancer types

## Graphical abstract

## Authors

Tae Yoon Park, Mark D.M. Leiserson, Gunnar W. Klau, Benjamin J. Raphael

## Correspondence
braphael@princeton.edu

## In brief

Using SuperDendrix, Park et al. examine associations between genetic dependencies in 769 cancer cell lines. They report 127 genetic dependencies explained by combinations of mutually exclusive somatic mutations congregating into a few oncogenic pathways across cancer subtypes. These present a small number of prominent and highly specific genetic vulnerabilities in cancer.

## Highlights

- SuperDendrix finds associations between sample features and CRISPR genetic dependencies

- Somatic mutations are associated with 127 genetic dependencies from Project DepMap

- Lineage-specific dependencies on transcription factors correlate with gene expression

- Identified associations agree with direction of interactions within oncogenic pathways

 Park et al., 2022, Cell Genomics 2, 100099
February 9, 2022 © 2022 The Author(s).

# Cell Genomics

CellPress

## Article

# SuperDendrix algorithm integrates genetic dependencies and genomic alterations across pathways and cancer types

Tae Yoon Park,[1,2,5] Mark D.M. Leiserson,[3,5] Gunnar W. Klau,[4,5] and Benjamin J. Raphael[1,2,6,*]
[1]Department of Computer Science, Princeton University, Princeton, NJ 08540, USA
[2]Lewis-Sigler Institute for Integrative Genomics, Princeton University, Princeton, NJ 08540, USA
[3]Department of Computer Science and Center for Bioinformatics and Computational Biology, University of Maryland, College Park, MD 20742, USA
[4]Algorithmic Bioinformatics, Heinrich Heine University Düsseldorf, Düsseldorf 40225, Germany
[5]These authors contributed equally
[6]Lead contact
*Correspondence: braphael@princeton.edu

## SUMMARY

Recent genome-wide CRISPR-Cas9 loss-of-function screens have identified genetic dependencies across many cancer cell lines. Associations between these dependencies and genomic alterations in the same cell lines reveal phenomena such as oncogene addiction and synthetic lethality. However, comprehensive identification of such associations is complicated by complex interactions between genes across genetically heterogeneous cancer types. We introduce and apply the algorithm SuperDendrix to CRISPR-Cas9 loss-of-function screens from 769 cancer cell lines, to identify differential dependencies across cell lines and to find associations between differential dependencies and combinations of genomic alterations and cell-type-specific markers. These associations respect the position and type of interactions within pathways: for example, we observe increased dependencies on downstream activators of pathways, such as *NFE2L2*, and decreased dependencies on upstream activators of pathways, such as *CDK6*. SuperDendrix also reveals dozens of dependencies on lineage-specific transcription factors, identifies cancer-type-specific correlations between dependencies, and enables annotation of individual mutated residues.

## INTRODUCTION

A key problem in cancer biology is to identify the genes that cancer cells depend on for their growth and survival advantage. This knowledge both informs our understanding of cancer development and suggests therapeutic targets.[5–7] Some cancer-essential genes are altered by somatic mutations and thus identified by high-throughput DNA sequencing,[8–10] but other cancer-essential genes are rarely or not mutated in cancer, such as lineage-specific transcription factors or master regulators.[11–14] An alternative approach to identify cancer-essential genes is to perturb genes in *in vitro* cancer models, such as cell lines, and measure growth or viability after such perturbations. Genes whose perturbation results in a change in viability reveal potential cancer-specific genetic dependencies. Recent technologies such as genome-wide pooled RNAi[15] or CRISPR[16,17] loss-of-function screens enable high-throughput genome-wide perturbation screens. Projects such as DRIVE[18] and the Cancer Dependency Map (DepMap)[19,20] have applied these technologies to hundreds of cancer cell lines and identified genes that are essential in specific cancer cell lines, often referred to as conditionally essential genes, or differential dependencies. Combining differential dependencies with genomic alterations identified in the same cell lines has revealed several context-specific dependencies including examples of oncogene addiction[21,22] and synthetic lethality.[23,24]

Several recent studies have attempted to systematically identify associations between differential dependencies and genomic alterations using data from large-scale RNAi and CRISPR datasets.[18–20,25–30] One group of studies identifies associations between differential dependencies and *single* genomic biomarkers,[18,20,26,27,29,30] recapitulating many of the classic oncogene addiction and synthetic lethal interactions, as well as a few additional associations. However, restriction to a single biomarker limits the ability to detect associations with rare genomic alterations that occur in a small number of cell lines but perturb the same cancer pathways as frequently mutated driver mutations.[8,9,31,32]

A second group of studies identifies associations between differential dependencies and sets of *multiple* biomarkers.[19,25,28] Tsherniak et al.[19] used a random forest classifier to predict dependencies in the DepMap dataset from genomic alterations. However, most of the thousands of reported associations were with gene expression markers and other frequent events, which

is not surprising since the classifier skews toward explaining the most frequent associations. REVEALER[25] and UNCOVER[28] leverage the observation that driver mutations in the same pathway tend to be mutually exclusive across tumors, i.e., that few tumors have more than one driver mutation in a given pathway.[33–35] These methods generalize earlier approaches that identify sets of mutually exclusive mutations in cancer genomes.[35–40] However, REVEALER does not scale to systematic analysis of large-scale screens,[28] while UNCOVER predicts hundreds to thousands of associations whose quality are generally unknown.

Spurious associations are a significant challenge in analyzing large-scale RNAi or CRISPR screens, since the number of phenotypes (gene perturbations) and number of features (genomic alterations) are orders of magnitude larger than the number of samples. This challenge is further exacerbated when searching for sets of multiple biomarkers as the number of such sets is massive and the optimal set is unknown *a priori*. Several related methods have also been developed to identify associations between genomic alterations and cancer dependencies measured from drug response experiments, including LOBICO,[41] CELLector,[42] and other methods using a penalized linear regression[43–47] and random forest.[47]

We introduce a new algorithm, SuperDendrix, to identify sets of approximately mutually exclusive genomic alteration and cell-type features that are associated with differential dependencies from large-scale perturbation experiments. SuperDendrix includes several key features: (1) a principled approach to identify and score differential dependencies using a 2-component mixture model; (2) a combinatorial model and optimization algorithm to find feature sets associated with differential dependencies; and (3) a model selection criterion to select the size of the associated set and a robust statistical test that accounts for different frequencies of genomic alterations across samples.

We apply SuperDendrix to identify associations between somatic mutations in cancer genes and differential dependencies in a large-scale CRISPR-Cas9 loss-of-function screens from DepMap[48] of 18,119 gene knockouts across 769 cancer cell lines from 31 cancer types. We identify 127 differential dependencies that are associated with sets of mutations. Many of these associations group into well-known cancer pathways including the NFE2L2, RB1, MAPK, and Wnt pathways. We observe that associations between differential dependencies and mutations within a pathway respect the topology and regulatory logic of the interactions within the pathway. Specifically, we find that cell lines containing oncogenic mutations in a gene upstream in a pathway—either activating mutations in an oncogene or inactivating mutations in a tumor suppressor gene—often have *increased dependencies* on genes downstream in the same pathway. These associations generalize the phenomenon of oncogene addiction to *oncogene pathway addiction*.[21,22] On the other hand, we find that oncogenic mutations in genes that are downstream in a pathway are often associated with *decreased dependency* on genes upstream in the same pathway. When including the cancer type as an additional feature for each cell line, SuperDendrix identifies a total of 227 differential dependencies that are associated with sets of mutations and/or cancer-type features, most prominently de-

pendencies on lineage-specific transcription factors and a previously unreported association between *TCF3* dependency and myeloma or blood cancers with mutations in *BCL2*.

The SuperDendrix software is publicly available at https://github.com/raphael-group/superdendrix and results on DepMap datasets are available through the web portal at https://superdendrix-explorer.lrgr.io/.

## RESULTS

### SuperDendrix

We introduce SuperDendrix, an algorithm to identify sets of binary features such as genomic alterations and/or cell types that are (approximately) mutually exclusive and associated with a quantitative phenotype. While SuperDendrix is applicable to any quantitative phenotype, in this work we focus on the phenotype of cell viability change following genome-wide CRISPR-Cas9 loss-of-function screens. The inputs to SuperDendrix are as follows.

> Cell viability measurements are from genome-wide CRISPR-Cas9 loss-of-function screens. We represent these measurements in a phenotype matrix $P$ where each entry $p_{gj}$ of $P$ indicates the viability of cell line $j$ when gene $g$ is knocked out. Each of these scores quantifies the *dependency* of a cell line on a gene. We refer to the dependency scores for a gene across cell lines (i.e., row of $P$) as a *dependency profile.* A list of somatic alterations in each cell line. Here, we analyze somatic missense and nonsense mutations.
> (Optionally) Categorical information (e.g., cell type) of each cell line.

SuperDendrix consists of three modules (Figure 1A): (1) scoring differential dependencies and selecting genomic and cell-type features; (2) finding feature sets associated with differential dependencies; and (3) evaluating the statistical significance of associations. We briefly describe the three modules below.

### Scoring differential dependencies and selecting genomic and cell-type features

The first module in SuperDendrix has two steps: (1) scoring differential dependencies from the dependency scores and (2) selecting the genomic alteration and cell-type features that will be evaluated for association. In the first step, we derive a *differential dependency profile* for each gene knockout (row of $P$). This profile quantifies the magnitude of the effect on the gene knockout on each cell line relative to a background distribution. We assume that the dependency scores $p_{g1}, \ldots, p_{gn}$ for knockout $g$ are generated from two populations: a background population that is unaffected by the knockout and a responsive population that is affected by the knockout. We fit a two-component mixture model to the dependency scores $p_{g1}, \ldots, p_{gn}$, and decide whether the score distribution is better fit by one-component or two-components using the Bayesian Information Criterion (BIC). In the case where the two-component fit is preferred, we say that the cell lines are differentially dependent with respect to the gene knockout $g$, or that gene $g$ is a differential dependency. We designate component 1 to be the component with smaller

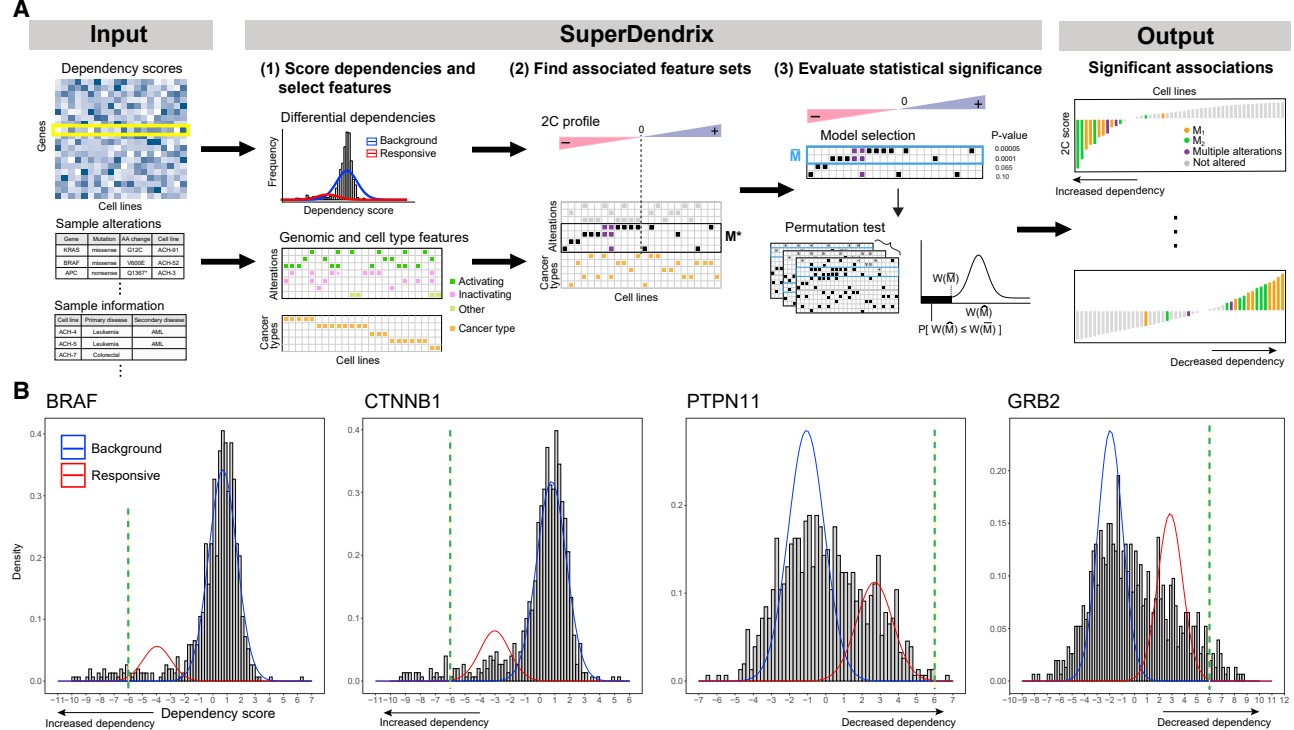

**Figure 1. Overview of SuperDendrix**

(A) SuperDendrix inputs are dependency scores of gene knockouts from CRISPR-Cas9 screens, genomic alterations, and optionally, cell-type features. In the first step, SuperDendrix derives differential dependencies—genes whose dependency scores are better fit by a mixture distribution of two components—and also constructs a genome alteration and cell-type feature matrix. In the second step, SuperDendrix finds a subset $M^*$ of features that maximize the SuperDendrix weight $W(M)$. In the third step, SuperDendrix performs model selection to define a subset $\overline{M}$ of features that substantially contribute to weight and computes statistical significance of weight $W(\overline{M})$ using a permutation test. Associations with false discovery rate (FDR) $\leq 0.2$ are output and include associations between features and increased dependency on profile (top right) and between features and decreased dependency on features (bottom right).

(B) Examples of differential dependencies from DepMap data that result from fitting the dependency scores with a mixture model. Blue curve is the background component, and red curve is the responsive component. Green dashed lines indicate $6\sigma$ criterion of Tsherniak et al.,[19] which identifies only a subset of cell lines that are responsive to knockout. *BRAF* and *CTNNB1* show increased dependency in response to knockout while *PTPN11* and *GRB2* show decreased dependency.

See also Figure S1.

mean and define the *differential dependency score*, or *2C score*,

$d_{gj} = \log \frac{\Pr(z_{gj} = 2 \mid p_{gj})}{\Pr(z_{gj} = 1 \mid p_{gj})}$ for cell line $j$ as the log ratio of the posterior

probabilities that cell line $j$ is from component 2 ($z_{gj} = 2$) and that cell line $j$ is from component 1 ($z_{gj} = 1$). Thus, negative 2C scores indicate decreased viability, or increased dependency in response to knockout. Conversely, positive 2C scores indicate decreased dependency in response to knockout. We assume that a minority of cell lines are responsive to gene knockout and thus refer to the component that contains fewer cell lines as the responsive component and the component with more cell lines as the background component. In summary, we say that differential dependencies whose responsive component has negative scores are increased dependencies and those whose responsive component has positive scores are decreased dependencies.

Next, we construct the genomic alteration and cell-type feature matrix $A$. This matrix contains two types of features. The first type are genomic alteration features. We define these features using the OncoKB database[49] to select genes and mu-tations with annotated roles in cancer. For each cancer gene in OncoKB, we use the functional annotations of non-synonymous somatic mutations to create three mutation features: activating mutations that confer gain-of-function are combined into a single feature labeled GENE(A), inactivating mutations that confer loss-of-function are combined into a single feature labeled GENE(I), and the remaining unannotated mutations are combined into a single feature labeled GENE(O). The second type of features in $A$ are cell-type features. In this analysis, we construct a binary feature for each cancer type represented in the analyzed cell lines. By definition, these cancer-type features are mutually exclusive across cell lines.

### Finding feature sets associated with differential dependencies

The second module in SuperDendrix is a rigorous and practically efficient combinatorial optimization algorithm to find sets $M$ of features in $A$ that are (1) approximately mutually exclusive and (2) associated with increased (or decreased) dependency. We derive the SuperDendrix weight $W(M)$ of a set $M$ that combines criteria (1) and (2) and use an integer linear

program (ILP) to find the set $M^*$ of minimum (or maximum) weight $W(M^*)$.

### Evaluating statistical significance of associations

The third module of SuperDendrix includes two steps. First, a model selection step identifies a subset $\overline{M}$ of the features in $M^*$ found in the second module, where each feature in $\overline{M}$ contributes significantly to the weight $W(\overline{M})$. This step uses a conditional permutation test to iteratively remove features whose contribution to the weight $W(M^*)$ is nearly the same as random features. Second, a permutation test assesses the statistical significance of the set $\overline{M}$. Since the number of somatic mutations varies considerably across cell lines (Figure S1), we use a permutation test[4] that conditions on both the number of mutations per gene and number of mutations per cell line.

We also developed an interactive tool for visualization and exploration of the SuperDendrix results which is available at https://superdendrix-explorer.lrgr.io/. Further details of the SuperDendrix algorithm are in STAR Methods.

### Identification of differential dependencies and genomic features in DepMap

We used SuperDendrix to analyze the Avana dataset (20Q2/5.20.2020) from Project DepMap containing results of CRISPR-Cas9 loss-of-function screens of 18,119 genes across 769 cancer cell lines from 31 cancer types.[19,48] DepMap provides a CERES dependency score[48] for each gene knockout across all cell lines. CERES scores are scaled across all gene knockouts so that the median score for known "essential" genes is −1 and the median score for genes with "no dependency" is 0. We define a set of differential dependencies from the CERES scores using the "6σ" criterion of Tsherniak et al.,[19] obtaining 2,074 genes that have at least one cell line with a CERES score at least six standard deviations below or above the mean. We refer to CERES score profiles for these 2,074 genes as 6σ differential dependencies (Table S1).

The first module of SuperDendrix computes that 511 (25%) of the 6σ differential dependencies are better fit by the two-component mixture model. We refer to these genes as two-component (2C) differential dependencies (Figure 1B, Table S2). These 511 2C differential dependencies include 446 genes with increased dependency and 65 genes with decreased dependency and are enriched for 108 GO molecular functions[50,51] including protein binding, enzyme binding, and catalytic activity (Table S3). Moreover, 88 of the 2C differential dependencies are in the COSMIC Cancer Gene Census (CGC)[52] (p ≤ 0.001)—including BRAF, KRAS, NRAS, and PIK3CA (Figure S2)—a significantly higher proportion than for non-2C genes (2C: 17.2%, non-2C: 11.2%, p ≤ 0.001; two-sample proportion test). In addition, the 2C differential dependencies include a significantly higher proportion of priority targets—differential dependencies identified based on gene knockout effect and biomarker correlation from CRISPR screens by the Sanger Institute[20]—than for non-2C genes (2C: 29.0%, non-2C: 13.4%, p ≤ 0.001; two-sample proportion test). The 2C differential dependencies have higher precision and recall for the priority targets than the differential dependencies identified by Normality Likelihood Ratio Test (NormLRT)[18] applied to the same dataset (see Comparison with NormLRT in STAR Methods). Finally, we find that the 6σ dif-

ferential dependencies that are not 2C differential dependencies either contain only a few outlier samples (e.g., 86.8% have fewer than 5 outlier samples) or have dependency score distributions that are unimodal with large variance (Figure S3).

We derive genomic features for SuperDendrix using 399,559 non-synonymous coding mutations reported in Cancer Cell Line Encyclopedia (CCLE)[44] for 767 of the 769 cell lines analyzed by DepMap. We also include the annotated cancer type of each cell line as a feature. The feature selection step in the first module of SuperDendrix produces a genomic alteration matrix with 897 mutation features (76 activating and 258 inactivating) in 621 genes with a total of 20,089 mutations across the 767 cell lines. Further details are in STAR Methods. We also evaluated SuperDendrix using different inputs including dependency probabilities provided by DepMap and all of the 399,559 non-synonymous mutations instead of the 897 mutation features obtained using OncoKB (see Analysis of CERES dependency probabilities and SuperDendrix analysis of all non-synonymous mutations in STAR Methods).

### Associations between mutations and differential dependencies

We used SuperDendrix to identify associations between sets of mutations and the 511 2C differential dependencies. SuperDendrix identified 91 single mutations and 36 sets of approximately mutually exclusive mutations that are significantly associated (FDR ≤ 0.2) with 127 differential dependencies (Figure 2A and Table S4). 113 of these sets are associated with increased dependency and 14 are associated with decreased dependency. Many of these associations are well-known dependencies including examples of oncogene addiction (e.g., BRAF(A) and increased dependency on BRAF[53]) and synthetic lethality (e.g., ARID1A(I) and increased dependency on ARID1B[54]). We find that the 127 genes with significant associations are enriched for 241 pathways annotated in the Reactome database.[55] Furthermore, 16 of the associations group into three well-known cancer pathways (NFE2L2, RB1, and MAPK). We highlight the novel findings of SuperDendrix in these pathways below.

First, SuperDendrix finds an association between the set {KEAP1(O), KEAP1(I), NFE2L2(A)} of three mutations and increased dependency on NFE2L2 (Figure 2B). The KEAP1-NFE2L2 pathway is frequently perturbed in cancer with inactivating mutations in KEAP1 or activating mutations in NFE2L2 reported in more than 30% of lung squamous tumors.[60,61] NFE2L2(A), KEAP1(I), or KEAP1(O) mutations occur in 69 of the 767 DepMap cell lines including 31% (5/16) of lung squamous cancer cell lines. Moreover, the three mutations are nearly mutually exclusive with only 3/69 altered cell lines having more than one mutation (Figure 2A). NFE2L2 is an oncogene in various cancers including lung, pancreas, breast, and gall bladder,[61,62] and thus increased dependency on NFE2L2 in cell lines with NFE2L2(A) mutations is consistent with the oncogene addiction model.[21,22] The increased NFE2L2 dependency in cell lines with KEAP1 inactivating mutations is consistent with KEAP1's role in inhibiting NFE2L2 by targeting NFE2L2 for degradation via ubiquitination.[56,63] Thus, the increased dependency on NFE2L2 in cell lines with KEAP1 inactivating mutations can be viewed as another form of oncogene addiction. These associations

**A**

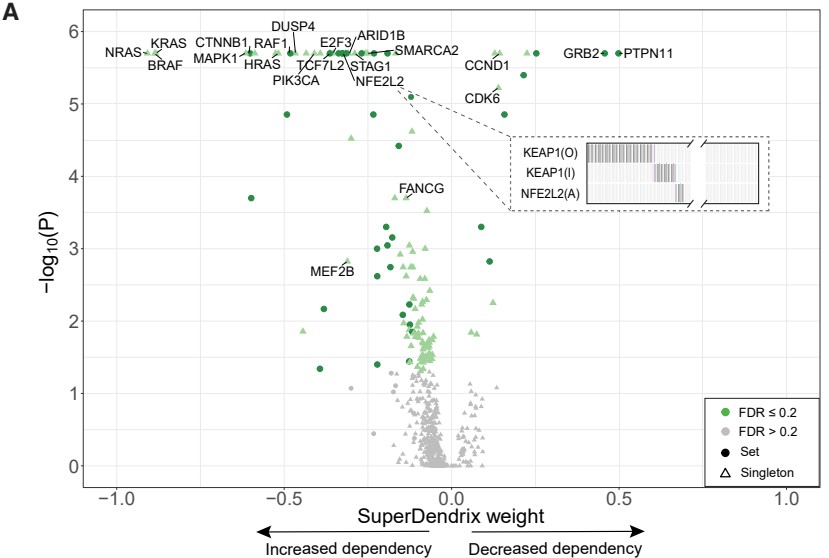

**B** NFE2L2

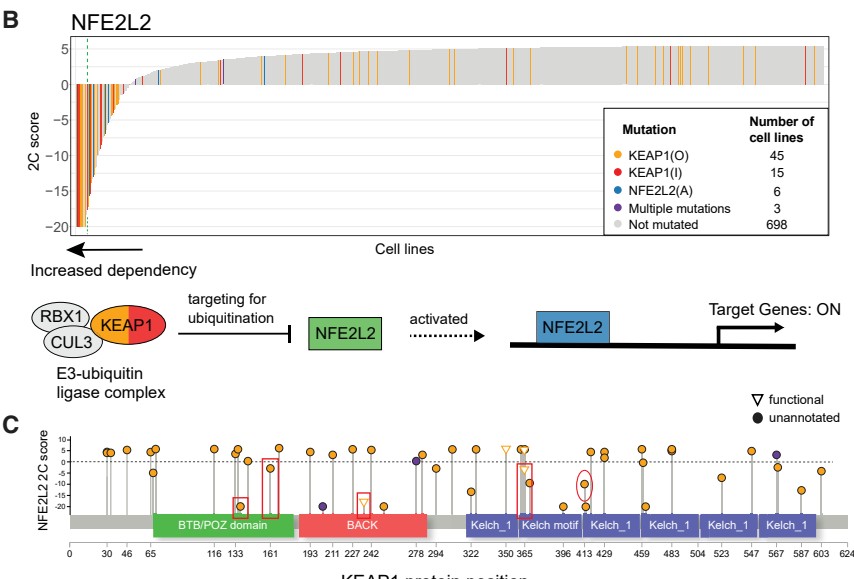

**Figure 2. SuperDendrix identifies associations between mutations and 2C differential dependencies in multiple biological pathways**

(A) SuperDendrix weights and p values for 127 2C differential dependencies with significant (FDR ≤ 0.2) associations with mutations. 36 of these associations are sets of multiple mutations; e.g., the set {KEAP1(O), KEAP1(I), NFE2L2(A)} are mutations that are approximately mutually exclusive and associated with increased dependency on *NFE2L2*.

(B) (Top) Waterfall plot of 2C differential dependency scores for *NFE2L2* across cell lines. Cell lines are colored by status in associated mutation set {KEAP1(O), KEAP1(I), NFE2L2(A)}. Green dashed line indicates 6σ threshold. (Bottom) KEAP1-NFE2L2 pathway. Solid circles are genes on the pathway, with colors indicating their mutations. Green boxes are genes that are knocked out. Association between *KEAP1* inactivating mutations and increased dependency on *NFE2L2* is consistent with the role of KEAP1 as an upstream activator of NFE2L2.

(C) Locations of missense mutations in KEAP1 protein that are annotated as other. KEAP1(O) mutations associated with increased dependency on *NFE2L2* include: two mutations in the BTB/POZ domain (boxed), a domain that is important for dimerization of KEAP1;[56] one annotated mutation in one of Kelch domains (boxed) which mediate interaction with NFE2L2;[57] and one mutation (circled) that lies at a residue that interfaces with NFE2L2.[58] Orange (resp. purple) amino acid changes are in cell lines with exclusive (resp. multiple) mutations in *KEAP1*. Triangles indicate locations of mutations that are reported in Uniprot[59] to affect KEAP1-NFE2L2 interaction.

See also Figures S2 and S3 and Tables S2, S3, and S4.

generalize the phenomenon of oncogene addiction to oncogenic pathway addiction:[21,22] mutations of genes in an oncogenic pathway confer strong dependency on other genes in the same pathway.

Note that only a fraction of the associated mutations occurs in cell lines whose CERES score is below the 6σ threshold (Figure 2B), demonstrating the advantages of SuperDendrix's 2C differential dependency score.

The associations with differential dependencies reported by SuperDendrix are also useful for annotating individual missense mutations. Specifically, several of the mutations in the KEAP1(O) feature—which include missense mutations that are unannotated in OncoKB—occur in cell lines with strong evidence of increased dependency on *NFE2L2*. For example, the *KEAP1* G364C mutation is not reported as functional in OncoKB, but

is located at a position that is reported to disrupt NFE2L2 repression.[57] Two other mutations are located in the BTB/POZ domain, a domain that is important for KEAP1 dimerization and KEAP1-CUL3 binding[56] (Figure 2C). Finally, the mutation R413L is located in Kelch domain and at the protein-protein interface with NFE2L2,[58] suggesting strong functional relevance of the mutation to KEAP1-NFE2L2 interaction. These findings prioritize these mutations for functional validation studies.

Second, SuperDendrix identifies associations between *RB1* inactivating mutations and differential dependencies in *E2F3*, *CCND1*, and *CDK6*, three members of the RB1 pathway (Figure 3). We find that cell lines with *RB1* inactivating mutations have increased dependency on *E2F3*. Active RB1 binds and inhibits E2F3 transcription factor activity, and dissociation of the RB1-E2F3 complex results in E2F3-initiated transcription of target genes that promote G1/S transition.[64] Our results suggest that cell lines with inactivating mutations in *RB1* become highly dependent on E2F3 activity, a phenomenon analogous to

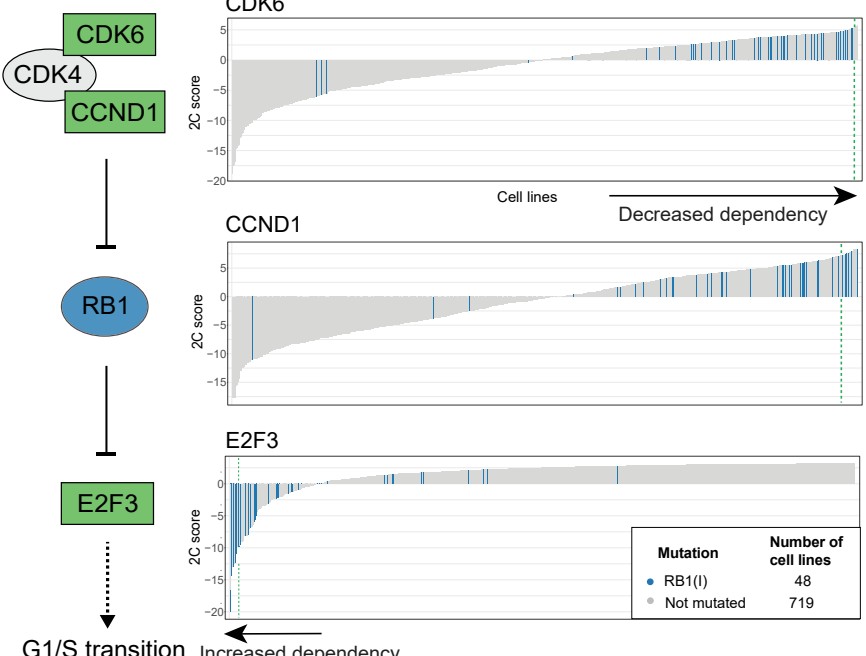

**Figure 3. SuperDendrix identifies associations between mutations and 2C differential dependencies in the RB1 pathway**

*RB1* inactivating mutations are associated with increased dependency on *E2F3*, consistent with RB1's role in inactivating the E2F3 transcription factor (same format as Figure 2B). On the other hand, *RB1* inactivating mutations are associated with decreased dependency on *CDK6* and *CCND1*. This is consistent with the role of the CDK4/6-CCND1 complex in inactivating RB1. See also Table S4.

oncogene addiction.[21,22] On the other hand, we observe that cell lines with *RB1* inactivating mutations are associated with decreased dependency on *CCND1* and on *CDK6*. This association is consistent with the role of CCND1-CDK4/6 complex in inactivating RB1. Cell lines with *RB1* inactivating mutations do not require CCND1 or CDK6 to inactivate RB1, making these cell lines *less* sensitive to knockout of CCND1 and CDK6. These results suggest a correspondence between the direction of dependencies and the patterns of activation/inactivation in a pathway. Similar to the oncogenic pathway addiction in the KEAP1-NFE2L2 pathway described above, we observe an increased dependency on the downstream transcription factor *E2F3* in cell lines with *RB1* inactivating mutations. On the other hand, we observe decreased dependency on the upstream regulators of *RB1*.

Third, SuperDendrix finds associations between 12 differential dependencies in the MAPK pathway and subsets of the approximately mutually exclusive mutation set {BRAF(A), KRAS(A), NRAS(A), HRAS(A)} (Figure 4A). These include well-known associations between activating mutations in *BRAF*, *KRAS*, *NRAS*, or *HRAS* and increased dependency on the corresponding gene.[53,65–67] Other associations involving *RAS* genes include an association between NRAS(A) and increased dependency on *SHOC2*,[68] as well as an association between the set {KRAS(A), NRAS(A)} of approximately mutually exclusive mutations and increased dependency on *RAF1*. The later association is consistent with the role of RAF1 as a mediator of RAS for signal transduction in the MAPK pathway during transformation.[69]

Also among associations identified in the MAPK pathway are associations between BRAF(A) mutations and increased dependencies on other downstream members of the MAPK signaling pathway including *MAP2K1*, *MAPK1*, *MITF*, and *DUSP4*. Associations with *MAP2K1*, *MAPK1*, and *MITF* are consistent with previous reports on conditional dependency on these genes in BRAF(V600E) melanoma.[70–72] The association with increased dependency on *DUSP4* is intriguing because there are conflicting reports regarding *DUSP4*'s role in cancer. On the one hand, *DUSP4* is reported to be a tumor suppressor that inhibits ERK1 and MAPK1 (ERK2) activity in the nucleus.[73,74] As a tumor suppressor, *DUSP4* knockout is expected to result in decreased dependency. On the other hand, there are also reports of high *DUSP4* expression in colorectal cancer[73,75] and skin cancer[76] with *RAS* or *RAF* mutations, suggesting that DUSP4 activity may contribute to oncogenesis in these cancers. Our finding that cell lines with BRAF(A) have increased dependency on *DUSP4* is consistent with the oncogenic role. To investigate these competing hypotheses, we investigated the relationship between *DUSP4* dependency and *MAPK1*, as *DUSP4* is a negative regulator of *MAPK1*. We found that in cell lines with BRAF(A), *DUSP4* dependency scores were significantly negatively correlated (R: −0.32, p ≤ 0.01; Pearson correlation) with expression of *MAPK1* (Figure 4B); i.e., cell lines with BRAF(A) and highest *MAPK* expression were the most dependent on *DUSP4*. In contrast, in cell lines without BRAF(A) mutations, there is no significant correlation between *DUSP4* dependency and *MAPK* expression (R: 0.01, p = 0.72). These observations are consistent with the Goldilocks principle[77] which states that precise levels of biological factors must be maintained for strong fitness, with either overdose or lack of oncogenic signal resulting in regression of tumor. In this case, *DUSP4* inhibition of *MAPK1* is most essential in cell lines with hyperactive MAPK signaling due to BRAF(A) mutations.

SuperDendrix also identifies associations between sets of mutations and decreased dependency on *PTPN11* and *GRB2* in the MAPK pathway. Specifically, we find decreased dependency on *PTPN11* in cell lines with KRAS(A), BRAF(A), or NRAS(A) mutations and decreased dependency on *GRB2* in the same cell lines. The decreased dependency on *PTPN11* is consistent with a previous report that cell lines with constitutive RAS or RAF signaling were insensitive to suppression of *PTPN11*.[78] While we are not aware of previous reports of associations with *GRB2*, it is intriguing that both proteins with decreased dependencies—PTPN11 and

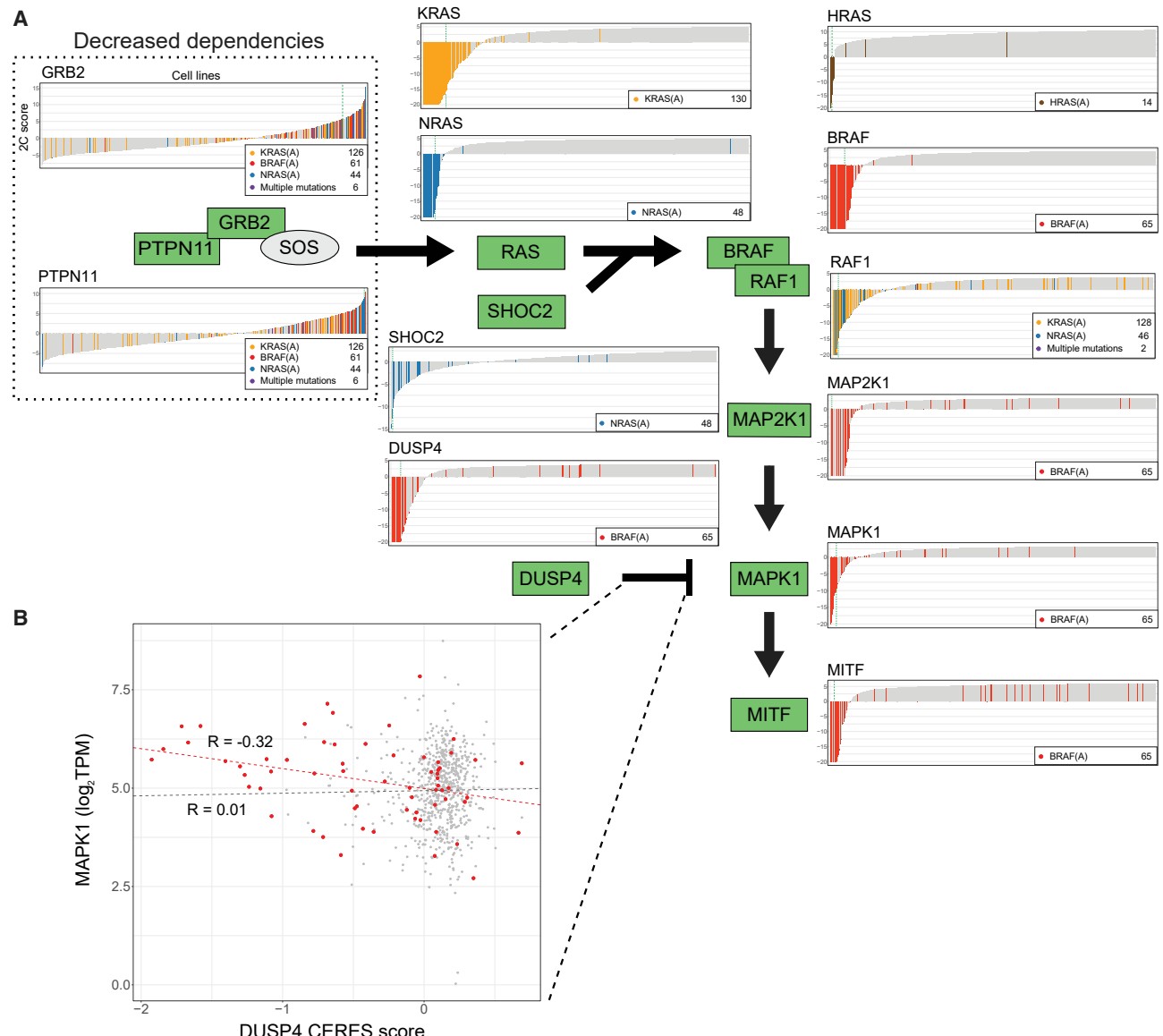

**Figure 4. Associations between mutations and 2C differential dependencies in the MAPK pathway**

(A) SuperDendrix identifies associations between approximately mutually exclusive activating mutations in *BRAF*, *KRAS*, *NRAS*, and *HRAS* and 12 differential dependencies in the MAPK pathway (same format as Figure 2B). Mutations that activate RAS/RAF are associated with increased dependencies of ten downstream genes in pathway. In contrast, these same mutations are associated with decreased dependency on two genes, *PTPN11* and *GRB2*, that are upstream activators of RAS.

(B) Expression of MAPK1 versus CERES dependency scores of DUSP4. Cell lines with activating mutations in *BRAF* (red dots) show negative correlation between *DUSP4* dependency score and *MAPK1* expression ($R = -0.32$, $p < 0.01$), while no correlation is observed in cell lines without *BRAF* activating mutations ($R = 0.01$, $p = 0.72$).

See also Table S4.

GRB2—are upstream of the RAS/RAF mutations that result in constitutive MAPK signaling. Thus, it makes sense that cell lines with constitutive activation of RAS or RAF signaling are insensitive to upstream activators of RAS signaling, analogous to the insensitivity of RB1-deficient cell lines to knockout of upstream regulators *CDK6* and *CCND1* reported above (Figure 3).

Beyond those in the three pathways described above, SuperDendrix identified other associations between members

of the same protein complex and associations in other cancer-implicated pathways. Associations in protein complexes include increased dependency on *ARID1B* in cell lines with ARID1A(I) mutation,[54] increased dependency on *SMARCA2* in cell lines with SMARCA4(I) mutation,[79] and increased dependency on *STAG1* in cell lines with STAG2(I) mutation.[80] Notable associations in pathways include associations in the Wnt pathway:[81] increased dependency on *CTNNB1* with the mutation set

{APC(I), CTNNB1(A)} and increased dependency on *TCF7L2* with the same set (Figure S4).

Several of the associations described above conform to the paradigm of oncogenic pathway addiction.[21,22] As a preliminary step for automatically identifying pathway addiction in a data-driven way, we performed a network analysis which integrates the associations identified by SuperDendrix with prior knowledge of physical interactions in a protein-protein interaction (PPI) network. This analysis identified a subnetwork (Figure S5) containing genes in multiple addicted pathways including the NFE2L2, MAPK, and Wnt pathways (see Network analysis of pathway addiction in STAR Methods for details).

We find that associations for 45 of the 127 differential dependencies reported by SuperDendrix are validated in the Score dataset of CRISPR screens from Behan et al.,[20] where we consider an association to be validated if there is a significant difference ($p \leq 0.05$; Wilcoxon rank sum test) in dependency scores between cell lines containing associated mutations and those without such mutations (Table S4). Many of the associations that did not validate are in cancer types with few cell lines in the Score dataset. For example, several associations with BRAF(A) did not validate in the Score dataset; this is not surprising since the majority of BRAF(A) mutations in the Avana dataset are in the 54 skin cancer cell lines, while the Score dataset contains only 4 skin cancer cell lines (Table S5). Further details are in Validation on the Sanger CRISPR-Cas9 screen data in STAR Methods.

Finally, we compared the associations between mutations and dependencies identified by SuperDendrix with those reported in other perturbation screens[19,20] (see Comparison with other perturbation screen results in STAR Methods) and to associations identified by other methods including a simple univariate test, UNCOVER,[28] and SELECT[40] (see Univariate analysis of the Dep-Map data, Comparison with UNCOVER, and Comparison with SELECT in STAR Methods). We found that SuperDendrix performed favorably in these comparisons.

### Cancer-type-specific differential dependencies

Next, we investigated associations between differential dependencies and cancer types. We augmented the mutation matrix with 31 cancer-type features, each feature representing one of the 31 cancer types in the Avana dataset. We ran SuperDendrix on the augmented mutation matrix and identified 298 differential dependencies that are significantly associated (FDR $\leq 0.2$) with mutations and/or cancer types (Table S6), with 227 of these including at least one cancer-type feature in the association. Among the 227 differential dependencies that are associated with at least one cancer-type feature are 68/127 differential dependencies that were identified in the SuperDendrix analysis of mutations described above. The sets associated with these differential dependencies include cancer-type features and result in higher SuperDendrix weights. For example, *MITF* dependency has stronger association with skin cancer (SuperDendrix weight = −0.69) than with BRAF(A) (SuperDendrix weight = −0.37).

Of the 227 differential dependencies, 195 are increased dependencies upon gene knockout and the other 32 are decreased dependencies. These 227 differential dependencies are en-

riched (FDR $\leq 0.05$) for 88 GO molecular function terms (Table S7). The most significant GO term is *protein domain specific binding*; in particular, 43 of the 227 differential dependencies are transcription factors[82] (fold enrichment = 2.19, $p \leq 0.001$), a greater proportion than the 67 transcription factors found among all 511 differential dependencies (fold enrichment = 1.51, $p \leq 0.001$). The enrichment of transcription factor dependencies is consistent with previous reports; e.g., Tsherniak et al.[19] identified 49 transcription factors with strong lineage-specific dependencies from RNAi screens. Our results include 16 of these 49 as well as 27 additional transcription factor dependencies that were not reported in the RNAi screens. As many transcription factors have lineage-specific expression, we evaluated the contribution of lineage and of transcription factor expression to the identified associations. We found that both the expression of the dependent transcription factor and the lineage classification are important for gene dependency across cell lines (Figure S6). Further details are in Expression of lineage-specific transcription factors in STAR Methods.

The 43 transcription factor dependencies with cancer-type-specific associations cluster into a number of interesting groups (Figure 5). These include increased dependencies on *ISL1*, *GATA3*, and *MYCN* in neuroblastoma, all of which were recently reported as part of the core regulatory circuitry (CRC) in neuroblastoma and associated with superenhancers.[83] We also find decreased dependencies on two transcription factors, *THAP1* and *TP53*, which are consistent with their functional role in the associated cancer types[84–87] (see Decreased dependency on transcription factors in STAR Methods). Other large classes of cancer-type dependencies are in skin cancer (6 dependencies), breast cancer (5), leukemia or lymphoma (9), and multiple myeloma (6).

Cancer-type-specific dependencies identified by SuperDendrix include associations between blood cancers (myeloma, lymphoma, and leukemia) and several transcription factors involved in B cell development[88] (Figure 5). Prominent examples are dependencies on transcription factors *TCF3* and *IRF4* which serve critical roles in determining B cell terminal differentiation into plasma cells (the cell type of myeloma cancer) via transcriptional regulatory activity[89–91] (Figure 6). Specifically, SuperDendrix finds associations between increased dependency on the transcription factor *TCF3* and the mutually exclusive set {myeloma, BCL2(A)} and the transcription factor *IRF4* and {myeloma, lymphoma}. SuperDendrix also finds associations for three downstream targets of TCF3 and IRF4 transcription factors: *BCL2* and {leukemia, myeloma, MEF2B(A)}, *PIM2* and myeloma, and *POU2AF1* and {Myeloma, MEF2B(A)}. The association between *BCL2* dependency and MEF2B(A) mutations is consistent with reports that MEF2B targets *BCL2* for transcriptional regulation.[92] Thus, this association conforms to the model of oncogenic pathway addiction, with increased dependency on *BCL2* in cell lines with activating mutations in the upstream transcriptional regulator *MEF2B* (analogous to the associations described above such as between increased dependency on *MAP2K1* and activating mutations in *BRAF*).

*TCF3* and *IRF4* have previously been suggested as dependencies in myeloma and are part of the core regulatory circuitry, promoting tumorigenesis in cooperation with aberrant MYC

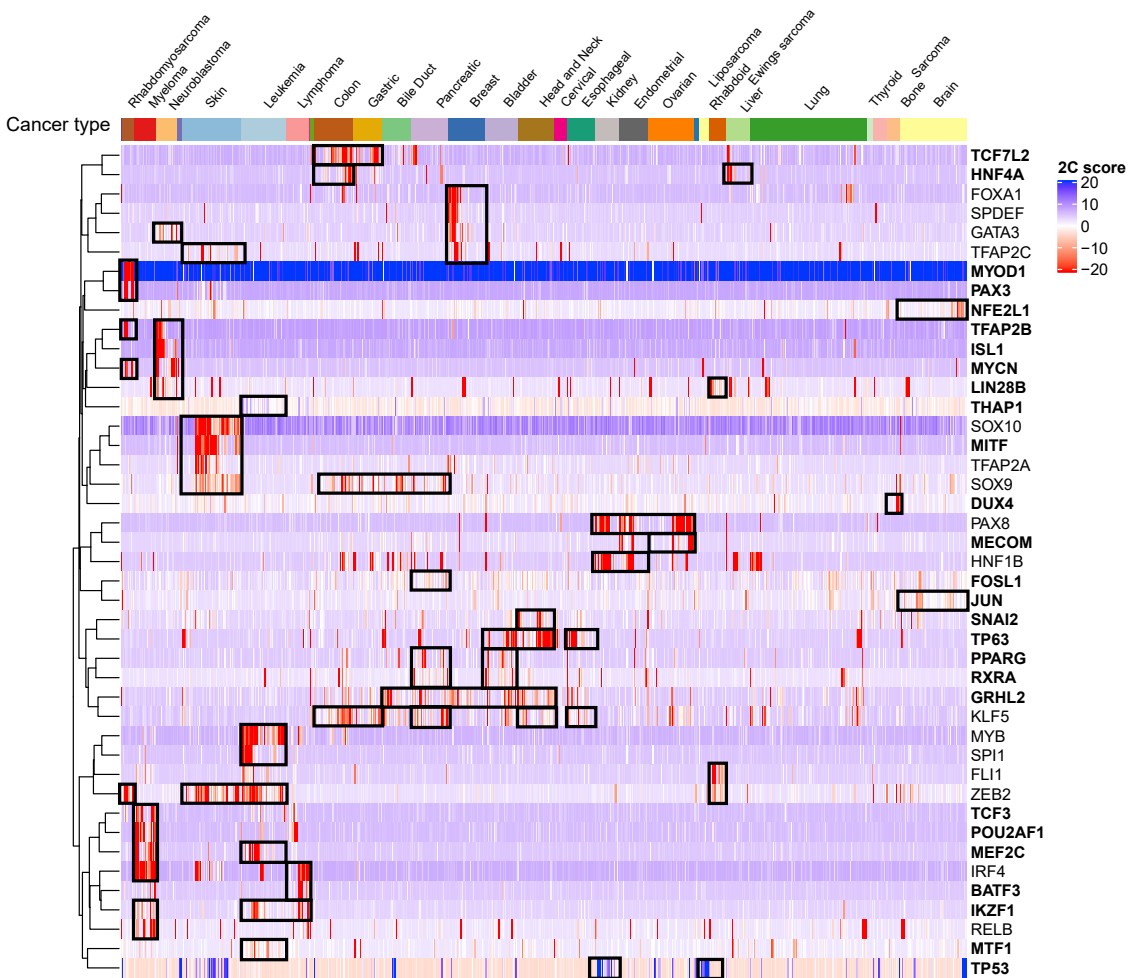

**Figure 5. Heatmap of 2C scores for 43 transcription factors identified by SuperDendrix as cancer-type-specific differential dependencies**
Dependency profiles are clustered within and across cancer types, with black boxes highlighting groups of prominent dependencies across cancer types. Bold text indicates transcription factors that were not reported in RNAi analysis.[19] Labels are shown for cancer types with at least 5 cell lines. See also Figure S6 and Tables S6, S7, and S8.

activity.[93] Consistent with these reports, *MYC* expression is higher in myeloma cell lines than other cancer types (p ≤ 0.001, Wilcoxon rank-sum test) and is significantly correlated with dependency on *TCF3* and *IRF4* (*TCF3*: R = −0.13, p ≤ 0.001, *IRF4*: R = −0.14, p ≤ 0.001; Pearson correlation, Figure S7). Dependencies on *POU2AF1* and *PIM2*, the target genes of TCF3 and IRF4 transcription factors,[91,94] suggest cancer-type-specific addiction to transcriptional regulatory pathway in myeloma. Supporting this notion, dependencies identified by SuperDendrix include other genes (e.g., *IKZF1*, *MEF2C*, *CCND2*) that are part of the regulatory network mediated by super-enhancer activity.[95,96] Taken together, these results show increased dependency on the B cell lineage-specific transcription factors in blood cancers, with cancer-type-specific addiction to TCF3/IRF4 regulatory pathway in myeloma mediated by *MYC* expression.

The remaining 184 cancer-type-specific differential dependencies that are not annotated as transcription factors are enriched (FDR ≤ 0.05) for 46 GO molecular function terms (Table S8), with the top 3 enriched terms being catalytic activity, ribonucleotide binding, and transferase activity, all of them transferring phosphorus-containing groups. These 184 differential dependencies include genes known to be overexpressed or predictive of prognosis for the associated cancer type such as increased dependencies on *LDB1* and *LMO2* in leukemia.[97,98] Several additional associations correspond to dependencies on upstream regulators of cancer genes such as *MDM2* in skin and kidney cancers and *EGFR* in head and neck cancer.

A prominent group of cancer-type-specific differential dependencies are six genes in the IGF1R and PI3K pathways (Figure 7A) across several cancer types. In the IGF1R pathway, we find increased dependency on *IGF2BP1*, *IGF1R*, *IRS1*, and *IRS2* in neuroblastoma, Ewing's sarcoma, pancreas, myeloma, or rhabdomyosarcoma. These dependencies are consistent with previous reports of dependencies on *IGF1R* in Ewing's sarcoma and rhabdomyosarcoma.[99] SuperDendrix also identifies

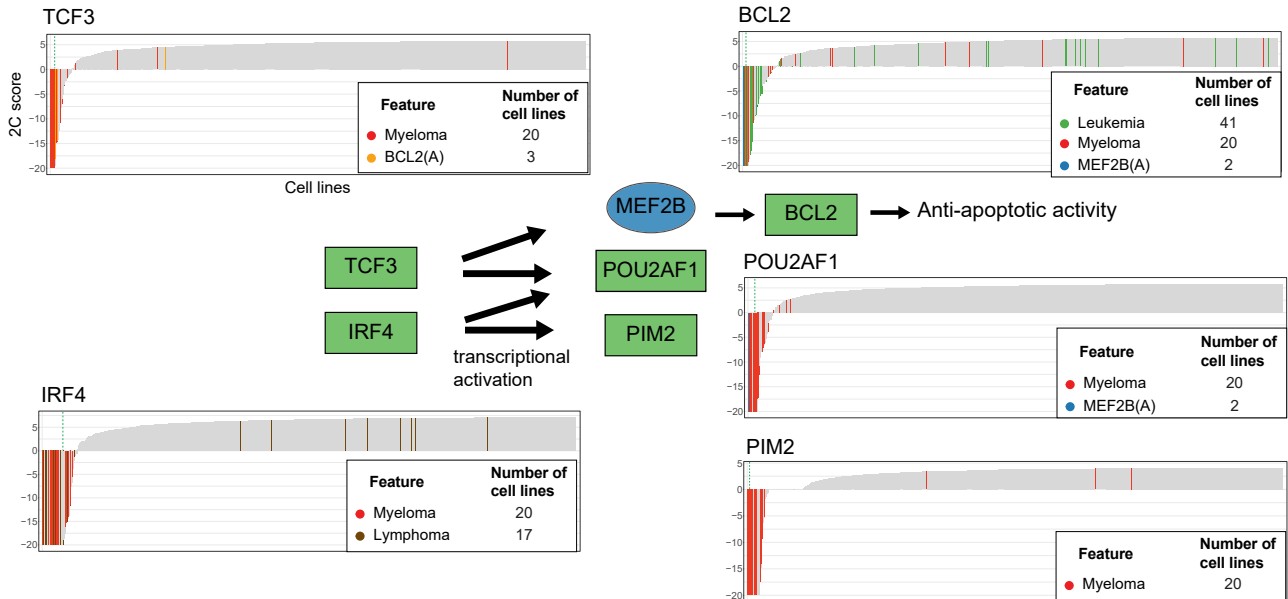

**Figure 6. Dependencies on TCF3 pathway genes in blood cancers**
SuperDendrix identifies cancer-type-specific dependencies on five genes of the TCF3 pathway in myeloma, leukemia, and lymphoma cell lines as well as cell lines with BCL2(A) and MEF2B(A) mutations. The five genes include two core regulatory transcription factors, *TCF3* and *IRF4*, and two genes regulated by these transcription factors. See also Figure S7 and Table S6.

increased dependencies on *PIK3CA* and *BCL2* in some of the same cancer types, including *PIK3CA* in myeloma and rhabdo-myosarcoma and *BCL2* in leukemia and myeloma.[100,101] These findings are consistent with role of IRS1 and IRS2 in activating the PI3K pathway.[102] Since dysregulation of the PI3K pathway results in tumor proliferation,[103] all of these increased dependencies are consistent with a phenotype of oncogenic pathway addiction in the IGF1R/PI3K pathway.

Additionally, we observed cancer-type-specific correlations between dependency scores of pairs of genes in the IGF1R pathway in neuroblastoma and Ewing's sarcoma. These include correlations between *IGF2BP1* and *IGF1R* ($R = 0.48$) in Ewing's sarcoma (Figure 7B) and between *IGF2BP1* and *IRS2* dependencies ($R = 0.32$) in Ewing's sarcoma and neuroblastoma (Figure 7C). Importantly, these correlations are weaker in other cancer types ($R = 0.11$ and $R = 0.06$, respectively) and conse-quently were not reported in two recent studies[27,104] that exam-ined correlations between dependency profiles across all cell lines in DepMap. In addition, many of the cell lines with these correlated dependencies have CERES scores larger than the −0.6 threshold used to define dependency in DepMap.[48] Thus, the identification of these correlations relied on both SuperDen-drix's 2C scores and SuperDendrix's ability to identify cancer-type-specific associations. At the same time, we find strong correlations between *IGF1R* with *IRS1* and *IGF1R* with *IRS2* across all cell lines, as previously reported.[27,104]

We find that 107 of the 227 associations identified by Super-Dendrix are validated in the Score dataset,[20] using the same test as described in the previous section (Table S6). Also as above, many of the associations that did not validate are in can-cer types that are not well represented in the Score dataset

including myeloma, skin, and rhabdomyosarcoma (Table S5). Further details are in Validation on the Sanger CRISPR-Cas9 screen data in STAR Methods.

Finally, we compared the associations identified by Super-Dendrix with those identified in the DepMap dataset using a simple univariate test and UNCOVER.[28] Analogous to the previ-ous analysis using mutation features only, we found that Super-Dendrix performed favorably in these comparisons. Further details are in Univariate analysis of cancer-type-specific differ-ential dependencies and Comparison with UNCOVER in STAR Methods.

**DISCUSSION**

We introduced SuperDendrix, a method that incorporates a prin-cipled statistical model and a practically efficient combinatorial algorithm for analyzing differential gene dependencies from perturbation experiments. SuperDendrix scores differential de-pendencies using a two-component mixture model and iden-tifies mutually exclusive sets of features—including genomic alterations and/or cancer types—that are associated with each differential dependency. Application of SuperDendrix to CRISPR-Cas9 loss-of-function screens in 769 cancer cell lines from Project DepMap revealed 511 differential dependencies and inferred associations between 127 (27.4%) of these depen-dencies and sets of somatic mutations in cancer genes. Many of these associations group into well-known cancer pathways such as NFE2L2, RB1, and MAPK. SuperDendrix reports that a higher fraction of differential dependencies are associated with muta-tions compared to previous analyses of RNAi and CRISPR screens.[19,20,30] This illustrates some of the advantages of the

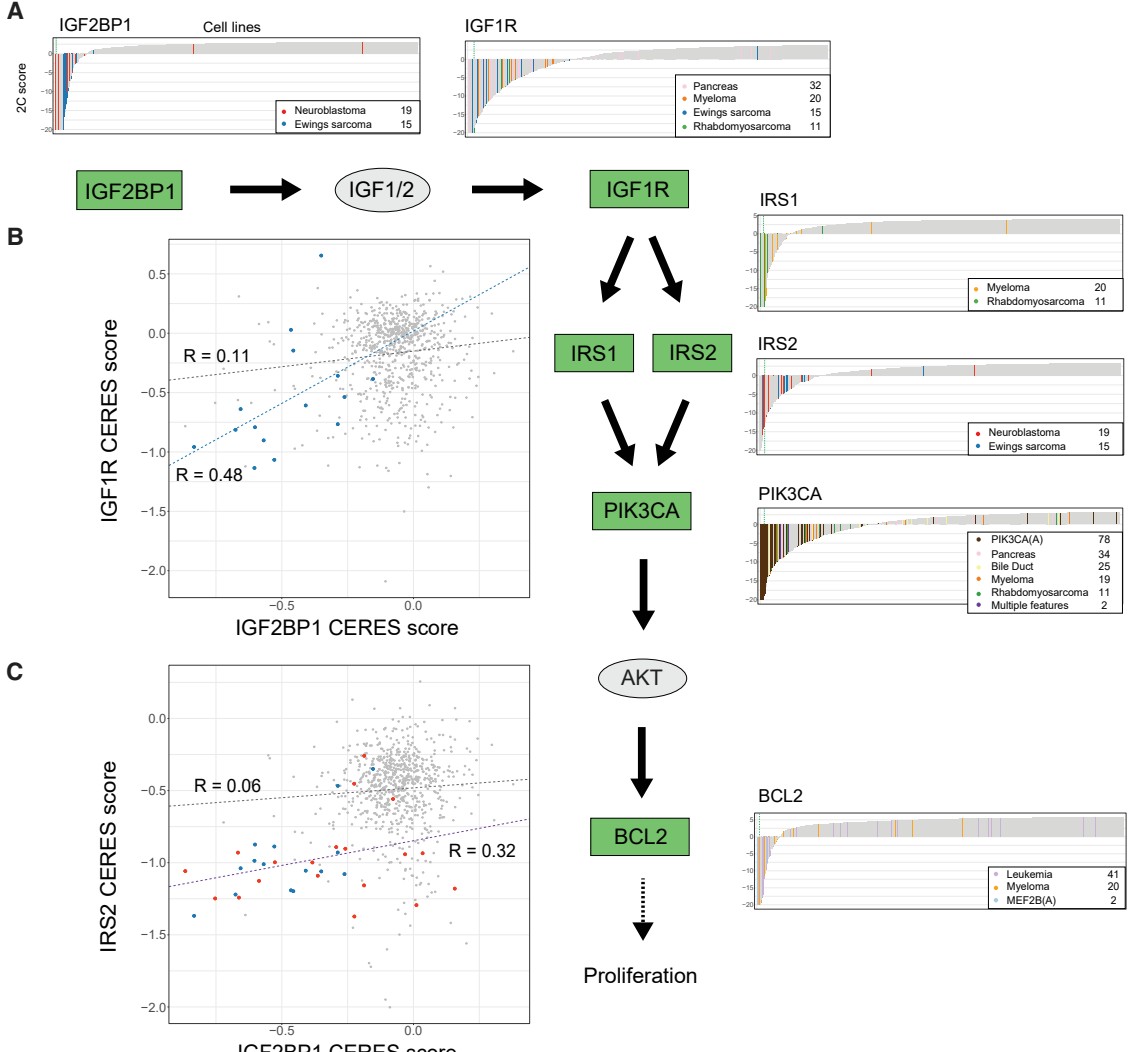

**Figure 7. Cancer-type-specific differential dependencies in the IGF1R/PI3K pathway**

(A) SuperDendrix identifies cancer-type-specific dependencies between six genes in IGF1R/PI3K pathway across multiple cancer types (same format as Figure 2B).

(B) CERES scores of *IGF2BP1* and *IGF1R* are positively correlated ($R = 0.48$) in Ewing's sarcoma cell lines (blue points) but only weakly correlated ($R = 0.11$) across other cancer types (gray points).

(C) CERES scores of *IGF2BP1* and *IRS2* are positively correlated ($R = 0.32$) in Ewing's sarcoma (blue points) and neuroblastoma (red points) cell lines, but only weakly correlated ($R = 0.06$) across other cancer types (gray points).

See also Table S6.

SuperDendrix method including more stringent selection of differential dependencies and searching for sets of associated biomarkers. In contrast, existing approaches relied on very permissive definitions of differential dependencies or restrict to finding associations with single biomarkers.

Our results show striking consistency between the directionality of dependencies (increased versus decreased), the type of interactions (activating versus inhibitory), and the position of dependencies and somatic mutations in pathways. In particular, oncogenic mutations in upstream pathway genes—such as activating mutations in an oncogene or inactivating mutations in a tumor suppressor—are associated with increased dependencies on genes that are downstream in the same pathway and that promote cancer; e.g., *NFE2L2* dependency in cell lines with inactivating mutations in *KEAP1* and *MAPK1* dependency in cell lines with activating mutations in *BRAF*. These results are consistent with the notion that cancer cells develop addiction to an oncogenic pathway during cancer progression.[21,22] On the other hand, oncogenic mutations in downstream pathway genes are associated with decreased dependencies on upstream genes of the same pathway; e.g., cell lines with inactivating mutations in *RB1* show decreased dependency on *CDK6*. These results show the importance of considering pathway topology in the design of cancer therapeutic strategies; for example, a current

strategy for treating tumors with activating mutations in undruggable oncogenes is to inactivate downstream genes.[105] At the same time, current annotations of interactions in pathways should be interpreted with care and potentially revised with knowledge gained from perturbation experiments. For example, *DUSP4* is noted as a tumor suppressor due to its role in inhibiting MAPK signaling; however, we find increased dependency on *DUSP4* in cell lines with activating mutations in *BRAF* suggesting that DUSP4 contributes to maintaining the balance of MAPK signaling in *BRAF* mutant tumors. These results suggest *DUSP4* as a potential therapeutic target for cancer treatment. Our results also provide further predictions about the functional consequences of individual non-synonymous mutations and the function of individual genes. For example, we find that previously unannotated mutations in the dimerization domain of *KEAP1* are associated with increased dependency on its downstream target *NFE2L2*.

SuperDendrix also identifies associations between differential dependencies and sets of cancer types or combinations of cancer types and somatic mutations. A large fraction (35%) of the cancer-type-specific associations found by SuperDendrix involve increased dependencies on lineage-specific transcription factors. Many of these lineage-specific transcription factors have been previously reported to be highly expressed or correlated with poor prognosis in cancers of corresponding types. We also identify associations that include both cancer types and somatic mutations. For example, we find that increased dependency on *BCL2* is associated with leukemia, myeloma, and *MEF2B* mutations. Another prominent cancer-type-specific association found by SuperDendrix is increased dependency on *IGF2BP1*, a regulator of insulin growth factor receptor *IGF1R*, in Ewing's sarcoma and neuroblastoma. We anticipate that with larger cohorts, there will be increased opportunities to identify these more subtle associations that include both cancer types and somatic mutations.

### Limitations of study

There are several limitations and caveats in the current study. First, all of the reported associations are computational predictions. While we strove for high specificity in these predictions, further experimental validation is warranted. Second, while we identified mutation and cancer type features that are associated with a large fraction of the differential dependencies, many of the differential dependencies remain unexplained due to either weak statistical significance or lack of associated cell line features. Some possible reasons for these unexplained differential dependencies are (1) the small sample size of 769 cell lines which limits statistical power to find rare associations particularly because the cell lines originate from a heterogeneous collection of 31 cancer types; (2) examination of a limited class of genomic alterations, namely non-synonymous single-nucleotide mutations; and (3) our modeling assumption that the mutations that are associated with a differential dependency are mutually exclusive. Including copy number aberrations (CNA) and DNA methylation changes in the analysis will likely identify additional associations; however, since these alterations span larger genomic distances than single-nucleotide mutations, they require more careful decomposition into spe-

cific alteration features.[38] Moreover, while the mutual exclusivity assumption is helpful for identifying combinations of mutations efficiently across hundreds of cell lines, there are reports of co-occurring driver mutations in cancer samples that cooperate to promote tumorigenesis. Thus, extending SuperDendrix to identify sets of co-occurring mutation features is an interesting future direction. Third, our identification of associations did not account for other covariates, although recent studies have demonstrated that CERES scores can be affected by other covariates and confounding variables such as tumor mutation burden, cell doubling time, cell cycle stage, growth media, culture type, etc.[58,106–108]

### Future directions

Beyond the limitations described above, there are several directions for future work, both the analysis and in further development of SuperDendrix. First, alternative dependency scores could be used as input to SuperDendrix.[109–111] Second, we found that some differential dependencies are associated with multiple sets of features (e.g., increased dependency on *TCF7L2* and the sets {APC(I), CTNNB1(A)} and {Colon,Gastric}). Extending SuperDendrix to simultaneously identify multiple sets of features might identify additional such dependencies, as previously shown for multiple sets of mutually exclusive mutations.[37] Third, one could integrate prior information on biological pathways to identify oncogenic pathway addiction in a data-driven way. Finally, since SuperDendrix is a general algorithm that can be used to find associations between binary features (e.g., germline or somatic mutations, cell types) and quantitative phenotypes (e.g., drug response, cell size), it would be interesting to analyze these other phenotypes using SuperDendrix, particularly drug response data from The Genomics of Drug Sensitivity in Cancer (GDSC) database,[43] and compare against other methods[41,112] that have been designed specifically to identify associations between drug response and genomic features.

### STAR★METHODS

Detailed methods are provided in the online version of this paper and include the following:

- KEY RESOURCES TABLE
- RESOURCE AVAILABILITY
  - Lead contact
  - Materials availability
  - Data and code availability
- METHOD DETAILS
  - SuperDendrix algorithm
  - Identifying differential dependencies and selecting genomic features
  - Finding feature sets associated with differential dependencies
  - Evaluating statistical significance of associations
- QUANTIFICATION AND STATISTICAL ANALYSIS
  - Bioinformatics and Data processing
  - Comparison with NormLRT
  - Analysis of CERES dependency probabilities

- ○ SuperDendrix analysis of all non-synonymous mutations
- ○ Network analysis of pathway addiction
- ○ Validation on the Sanger CRISPR-Cas9 screen data
- ○ Comparison with other perturbation screen results
- ○ Univariate analysis of the DepMap data
- ○ Univariate analysis of cancer-type-specific differential dependencies
- ○ Comparison with UNCOVER
- ○ Comparison with SELECT
- ○ Expression of lineage-specific transcription factors
- ○ Decreased dependency on transcription factors
- ● ADDITIONAL RESOURCES
  - ○ Web browser for genetic dependency and mutation data

## SUPPLEMENTAL INFORMATION

## ACKNOWLEDGMENTS

G.W.K. thanks the Fulbright Scholarship Program for support. This work is supported by US National Institutes of Health (NIH) grants R01HG007069 (B.J.R.), U24CA211000 (B.J.R.), and U24CA264027 (B.J.R.) and the Princeton Catalysis Initiative.

## AUTHOR CONTRIBUTIONS

Conceptualization, M.D.M.L., G.W.K., and B.J.R.; Software, T.P., M.D.M.L., and G.W.K.; Validation, T.P. and B.J.R.; Investigation, T.P., M.D.M.L., G.W.K., and B.J.R.; Writing – Original Draft, T.P., M.D.M.L., G.W.K., and B.J.R.; Writing – Review & Editing, T.P. and B.J.R.; Visualization, T.P., M.D.M.L., and G.W.K.; Funding Acquisition, G.W.K. and B.J.R.; Supervision, B.J.R.

## DECLARATION OF INTERESTS

B.J.R. is a cofounder of, and consultant to, Medley Genomics.

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

## STAR★METHODS

### KEY RESOURCES TABLE

| REAGENT or RESOURCE | SOURCE | IDENTIFIER |
| --- | --- | --- |
| **Deposited Data** | | |
| DepMap data (20Q2 release) | Meyers et al.[48] | https://depmap.org/portal/download/ |
| Project Score data (Release 1) | Behan et al.[20] | https://score.depmap.sanger.ac.uk/downloads |
| Cell Model Passports | van der Meer et al.[113] | https://cellmodelpassports.sanger.ac.uk/downloads |
| HINT+HI interaction network | Reyna et al.[114] | https://github.com/raphael-group/netmix |
| **Software and Algorithms** | | |
| SuperDendrix | This paper | https://github.com/raphael-group/superdendrix, https://doi.org/10.5281/zenodo.5885806 |
| SuperDendrix web browser | This paper | https://github.com/lrgr/superdendrix-explorer, https://doi.org/10.5281/zenodo.5878914 |
| Oncokb-annotator | Chakravarty et al.[49] | https://github.com/oncokb/oncokb-annotator |
| Gurobi | Gurobi Optimization, LLC[3] | https://www.gurobi.com |
| EMMIXskew | Wang et al.[1] | https://cran.r-project.org/src/contrib/Archive/EMMIXskew/ |
| Scikit-learn | Pedgregosa et al.[2] | https://scikit-learn.org/stable/ |
| Curveball | Strona et al.[4] | https://doi.org/10.1038/ncomms5114 |
| UNCOVER | Basso et al.[28] | https://github.com/VandinLab/UNCOVER |
| SELECT | Mina et al.[40] | http://ciriellolab.org/select/select.html |

### RESOURCE AVAILABILITY

#### Lead contact

Further information and requests for resources and data should be directed to and will be fulfilled by the Lead Contact, Benjamin J. Raphael (braphael@princeton.edu)

#### Materials availability

This study did not generate new reagents.

#### Data and code availability

This paper analyzes existing, publicly available data. The datasets are listed in the Key resources table.

We implement SuperDendrix using Python 3 and R. We use oncokb-annotator to annotate mutations. We use the R package, EMMIXskew,[1] to fit $t$-distribution mixture models to dependency scores. We use the Python scikit-learn library[2] to fit Gaussian mixture models to dependency scores and to compute the 2C scores. We use the Gurobi software[3] to solve the ILP in SuperDendrix and the Curveball software[4] to conduct permutation test. SuperDendrix software is publicly available at https://github.com/raphael-group/superdendrix (Zenodo: https://doi.org/10.5281/zenodo.5885806).

Any additional information required to reanalyze the data reported in this paper is available from the lead contact upon request.

### METHOD DETAILS

#### SuperDendrix algorithm

We introduce a new algorithm, SuperDendrix, to identify sets of binary features (e.g., genomic alterations or cell types) that are approximately mutually exclusive and associated with a continuous-valued phenotype. The inputs to SuperDendrix are:

1. An $l \times n$ matrix $P = [p_{gj}]$ of $l$ quantitative phenotypes measured in $n$ samples. Each entry $p_{gj}$ is the score of phenotype $g$ in sample $j$. Each row of the phenotype matrix corresponds to a phenotype profile.

 **CellPress**

**Cell Genomics**
Article

2. A list of binary features (e.g., somatic mutations) for each sample.
3. (Optional) Categorical information (e.g., cell type) of each sample.

While SuperDendrix is a general-purpose algorithm, here we describe the specific application where the phenotype scores are dependency scores from gene perturbation experiments and the binary features are somatic mutations (and optionally cell types). SuperDendrix includes three modules: (1) A module to identify and score differential dependencies using a two-component mixture model and to select genomic and cell-type features using mutation annotations; (2) A module to find sets of features that are approximately mutually exclusive and associated with differential dependencies using a combinatorial optimization algorithm; (3) A module to perform model selection and to evaluate statistical significance of associations.

### Identifying differential dependencies and selecting genomic features

The first module in SuperDendrix includes two steps: the identification and scoring of differential dependencies and the selection of genomic and cell-type features. In the first step, we assume that a gene perturbation leads to two population of samples: a minority of samples are *responsive* to the perturbation while the remaining samples are unresponsive and have scores derived from a *background* distribution. Thus, we assume that the dependency scores are distributed according to a two-component mixture model. We fit each dependency profile with a *t*-distribution and with a mixture of two *t*-distributions, using the *t*-distribution to model high variance in the dependency scores.[115] We use the Bayesian information criterion (BIC)[116] to select between the one-component or two-component models; we refer to genes whose dependency profiles are better fit by a two-component mixture as *differential dependencies*.

For each differential dependency $g$ and sample $j$, we compute the 2C score, or differential dependency score, $d_{gj} = \frac{\Pr(z_{gj}=2|p_{gj})}{\Pr(z_{gj}=1|p_{gj})}$, the log ratio of the posterior probabilities that the observed score is from component 2 or component 1. We compute posterior probabilities by fitting the dependency scores to a mixture of two *Gaussian distributions*. We choose component 1 to be the component with smaller mean so that negative 2C scores indicate decreased viability, or *increased dependency* in response to knockout. Conversely, positive 2C scores indicate *decreased dependency* in response to knockout. We assume that a minority of cell lines are responsive to gene knockout and thus refer to the component that contains fewer cell lines as the *responsive* component and the component with more cells lines as the *background* component. We define the 2C profile, or differential dependency profile, $d_g = (d_{g1}, \ldots, d_{gn})$ to be the differential dependency scores across all samples. Profile $d_g$ is an *increased dependency* if its responsive component contains cell lines with negative 2C scores (increased dependency) and a *decreased dependency* if its responsive component contains cell lines with positive 2C scores (decreased dependency).

In the second step, we construct a genomic alteration and cell-type feature matrix $A$ that includes annotated mutations and (optionally) cell-type features. We construct $A$ from non-synonymous somatic mutations in cancer genes in the OncoKB database.[49] We first annotate the input list of somatic mutations using the oncokb-annotator. This adds information on whether the gene has been curated in OncoKB (GENE_IN_ONCOKB), ability to induce cancer (ONCOGENIC), and biological effect (MUTATION_EFFECT) to each mutation.

For each GENE in OncoKB, we group mutations from the input list into **A**ctivating, **I**nactivating, or **O**ther mutation features which we label as GENE(A), GENE(I), and GENE(O) using the OncoKB annotation according to the following rules:

1. Mutations that are not oncogenic (Likely Neutral, Inconclusive Unknown) are grouped into a feature, GENE(O).
2. Oncogenic mutations (Oncogenic, Likely Oncogenic) with Gain-of-function or Likely Gain-of-function effect are grouped into a feature, GENE(A).
3. Oncogenic mutations with Loss-of-function or Likely Loss-of-function effect are grouped into a feature, GENE(I).
4. Oncogenic mutations with other effects are grouped into a feature, GENE(O).

Using the OncoKB mutation features derived above, we construct the feature matrix $A = [a_{ij}]$ of $m$ OncoKB mutation features across $n$ samples where $a_{ij} = 1$ if mutation $i$ occurs in sample $j$ and $a_{ij} = 0$ otherwise.

Next, we generate binary features that represent the cell type of each sample using information from metadata such as the primary tissue. In the application in this paper, we use cancer types as the cell-type features. Each cancer-type feature has the value 1 for samples of the corresponding cancer type and the value 0 for samples of other cancer types. Note that the cancer-type features are mutually exclusive by definition.

We now combine the two sets of features and create an augmented binary feature matrix $A$ of $m$ OncoKB mutation features and $q$ cancer-type features across $n$ samples.

### Finding feature sets associated with differential dependencies

The second module in SuperDendrix finds a subset $M^*$ of features (rows in $A$) that are: (i) most associated with differential dependency profile $d_g$; and (ii) approximately mutually exclusive.

First, for each score $d_{gj}$ of differential dependency $g$ in sample $j$ from the profile $d_g$, we define a normalized score $dI_j = \frac{d_{gj}}{S}$ where $S = \sum_{d_{gj}<0} d_{gj}$ if $d_g$ is an increased dependency and $S = \sum_{d_{gj}>0} d_{gj}$ if $d_g$ is a decreased dependency. Then, we define a weight function $W(M)$

that quantifies how well a subset $M = (m_1, \ldots, m_k)$ of features satisfies properties (i) and (ii). For the weight function $W(M)$, we generalize the weight function defined previously[35] to measure the mutual exclusivity between mutations. Specifically, for a set $M$, let $\Gamma(M)$ be the subset of samples with mutations in $M$, $c_j(M)$ be the number of mutations in $M$ that occur in sample $j$, and $\rho_j$ be a penalty term for mutations in $M$ that co-occur in sample $j$. When searching for association to increased dependency, $\rho_j$ is equal to $-\left|d'_j\right|$; when searching for association to decreased dependency, $\rho_j$ is equal to $\left|d'_j\right|$. We define

$$W(M) = \sum_{j \in \Gamma(M)} d'_j - (c_j(M) - 1)\rho_j. \qquad \text{(Equation 1)}$$

If the mutations in $M$ are mutually exclusive, then $c_j(M) = 1$ for all $j$ and thus $W(M)$ is the sum of differential dependency scores for all altered samples. If $c_j(M) > 1$, then sample $j$ has mutations in more than one feature in $M$, and thus we penalize the weight $W(M)$ for each additional mutation. Note that if the features that co-occur in a sample are GENE(I) and GENE(O) mutations, we do not penalize the weight. This is motivated by the two-hit hypothesis[117] which states that both alleles need to be mutated for gene inactivation. To see that the weight $W(M)$ is a straightforward generalization of the Dendrix weight introduced previously[35] we consider the following reformulation, in which $\Gamma(m)$ denotes the set of samples with feature $m$.

$$W(M) = \sum_{j \in \Gamma(M)} d'_j - (c_j(M) - 1)\rho_j = \sum_{j \in \Gamma(M)} d'_j + \rho_j - c_j(M)\rho_j$$

$$= \sum_{j \in \Gamma(M)} \left(d'_j + \rho_j\right) - \sum_{m \in M} \sum_{j \in \Gamma(m)} \rho_j$$

In the case where all samples have equal score, i.e., $d'_j = 1$, and $\rho_j = \left|d'_j\right| = 1$ for all $j$, the supervised Dendrix weight $W(M)$ simplifies to $W(M) = 2|\Gamma(M)| - \sum_{m \in M}|\Gamma(m)|$, which is the original Dendrix weight.[35]

Following the nomenclature in machine learning, the problem considered in Dendrix[35] of finding a mutually exclusive set of alterations is an "unsupervised" feature selection problem, while the problem solved by SuperDendrix is a "supervised" feature selection problem where we aim to identify a set of mutually exclusive features that "explain" a phenotype.

Next, we aim to find a set $M^*$ with optimal weight $W(M^*)$, which we define as follows.

*Problem 1* (Optimal Weight Exclusive Target Coverage Problem (OWXTC)). Given a binary feature matrix $A$ and a differential dependency profile $d$, find a subset $M^*$ of rows satisfying

$$M^* = \begin{cases} \arg\min_{M \subseteq \mathcal{E}} W(M) & \textit{for increased dependencies}, \\ \arg\max_{M \subseteq \mathcal{E}} W(M) & \textit{for decreased dependencies}, \end{cases} \qquad \text{(Equation 2)}$$

where $\mathcal{E}$ is all subsets of rows in $A$.

OWXTC is NP-hard because it generalizes the Maximum Weight Submatrix Problem which was shown to be NP-hard[35] for the special case where $d'_j = 1$, and $\rho_j = 1$ for all $j$. We also define the cardinality-constrained version $k$-OWTXC of OWXTC in which $\mathcal{E}$ is all subsets of size at most $k$.

We formulate the OWXTC as an integer linear program (ILP) as follows. First, we define binary variables $x_i$, for each row $1 \leq i \leq m$, and $y_j$, for each column $1 \leq j \leq n$, with the interpretation

$$x_i = \begin{cases} 1 & i \in M^* \\ 0 & \textit{otherwise} \end{cases} \textit{and} \; y_j = \begin{cases} 1 & a_{ij} = 1 \textit{ for some } i \in M^* \\ 0 & \textit{otherwise}. \end{cases}$$

Then the OWXTC in the case of increased dependency is equivalent to the following ILP.

$$\min \sum_{d'_j < 0} \left(d'_j + \rho_j\right) y_j - \sum_i \sum_{j \in \Gamma(i)} \rho_j x_i \qquad \text{(Equation 3)}$$

$$\text{subject to } y_j \leq \sum_{i : a_{ij} = 1} x_i \quad \textit{for all } 1 \leq j \leq n \qquad \text{(Equation 4)}$$

$$y_j \geq x_i \quad \textit{for all } i, j : a_{ij} = 1 \textit{ and } w_j < 0 \qquad \text{(Equation 5)}$$

$$x_i \in \{0, 1\} \quad \textit{for all } 1 \le i \le m \tag{Equation 6}$$

$$y_j \in \{0, 1\} \quad \textit{for all } 1 \le j \le n \tag{Equation 7}$$

For finding associations with decreased dependencies, we replace min by max in Equation (3). For the cardinality-restricted version, we add the inequality

$$\sum_{i \in \mathcal{E}} x_i \le k \tag{Equation 8}$$

Note that the SuperDendrix weight and the ILP are similar, but not identical, to those presented previously.[28] The differences are discussed in "Comparison with UNCOVER."

### Evaluating statistical significance of associations

The third module of SuperDendrix consists of two steps. First, since the optimal size $k = |M^*|$ of the feature set is unknown, we perform model selection using a conditional permutation test to evaluate the contribution of each mutation to the weight $W(M^*)$. For each feature $m$ in $M^*$, we compare the weight $W(M^*)$ to the distribution of the weight $W(\overline{M}^*)$, where $W(\overline{M}^*)$ is the weight obtained when mutations of the feature $m$ are permuted across samples. We compute the empirical $P$-value as $p_m = \Pr[W(\overline{M}^*) \le W(M^*)]$ (increased dependency) or $p_m = \Pr[W(\overline{M}^*) \ge W(M^*)]$ (decreased dependency) over 10,000 permutations and remove $m$ with the largest $P$-value only if $p_m > 0.0001$. We repeat the above process until we obtain a feature set $\overline{M}$ which only contains features with $p_m \le 0.0001$.

Next, we evaluate the statistical significance of the association between feature set $\overline{M}$ and differential dependency profile $d_g$ by running SuperDendrix on random feature matrices $\widehat{A}$ with fixed row and column sums[4] (numbers of mutations per gene and sample, respectively). Note that we generate these random matrices using *all* mutations (i.e., including mutations not annotated in OncoKB), and then use the first module in SuperDendrix to select the OncoKB mutation features. We compare the weight $W(\overline{M})$ to the distribution of the weight $W(\widehat{M})$, where $W(\widehat{M})$ is the optimal weight computed from a random feature matrix $\widehat{A}$. We compute the empirical $P$-value as $p = \Pr[W(\widehat{M}) \le W(\overline{M})]$ (increased dependency) or $p = \Pr[W(\widehat{M}) \ge W(\overline{M})]$ (decreased dependency) over up to 500,000 random feature matrices. After computing $P$-values of the feature sets for each differential dependency, we compute false discovery rate (FDR) using Benjamini-Hochberg procedure[118] for multiple hypothesis correction.

## QUANTIFICATION AND STATISTICAL ANALYSIS

### Bioinformatics and Data processing

We downloaded the Avana [20Q2/5.20.2020] dataset[48] from the DepMap data portal. This dataset contains dependency scores – computed using the CERES algorithm – for 18,119 CRISPR-Cas9 gene knockouts across 769 cancer cell lines. We normalize each of 18,119 dependency profiles by converting CERES scores to z-scores as described in Meyers et al.[48] before applying the first module of SuperDendrix. After running the first module of SuperDendrix, we obtain 511 differential dependencies that are better fit by a mixture of two *t*-distributions; 446 increased dependencies and 65 decreased dependencies.

We downloaded non-synonymous somatic mutation data [20Q2/5.20.2020] for the same cell lines from the Cancer Cell Line Encyclopedia (CCLE)[44] using the same DepMap data portal. This dataset includes 547,597 mutation data for 767 of the 769 cell lines in the CRISPR-Cas9 dataset. We excluded "silent" and "other conserving" mutations and applied SuperDendrix to 399,559 non-synonymous mutations. After running the first module of SuperDendrix we obtain a genomic alteration matrix containing 897 mutation features (76 GENE(A), 258 GENE(I), and 563 GENE(O) mutation features) in 355 genes in 767 cell lines. Note that these mutation features do not overlap with the list of recently identified "passenger hotspot" mutations caused by preferential APOBEC activity in DNA stem loops.[119] To derive cancer-type features, we used the "primary_disease" and "Subtype" columns in the DepMap cell line metadata and fixed annotation errors and merged rare cancer sub-types. Our annotation of cancer types is in "Cancer_type" column in our curated cell line data (Table S9). We use this annotation to construct 31 binary cancer-type features representing the cancer types of DepMap cell lines where each feature has a value 1 for cell lines of that cancer type and 0 for other cell lines.

We run SuperDendrix using sets of at most 3 mutations and sets of at most 5 mutations and/or cancer types.

### Comparison with NormLRT

We compare two sets of differential dependencies identified from the DepMap data using the two-component mixture model from SuperDendrix and Normality Likelihood Ratio Test (NormLRT).[18,30] NormLRT measures the divergence of a dependency profile from a Gaussian distribution by fitting the dependency scores to a Gaussian and a skewed-t distribution. LRT score is the following:

$$LRT = 2 \times [\ln(\textit{likelihood for Skewed} - t) - \ln(\textit{likelihood for Gaussian})]$$

SuperDendrix identified 511 2C differential dependencies while NormLRT identified 949 differential dependencies using the same LRT score threshold of 125 from the original study.[18,30] We compare the two lists of differential dependencies to reference gene sets of Sanger priority target genes[20] and nonessential genes[111] that were identified based on gene dependency from independent CRISPR screens.

SuperDendrix outperforms NormLRT in identifying known dependencies, achieving both higher precision and recall for Sanger priority targets (SuperDendrix: 0.1, NormLRT: 0.03; area under the precision-recall curve (AUPRC)). In addition, differential dependencies from SuperDendrix contain fewer nonessential genes than NormLRT differential dependencies (NormLRT: 3.2% (30/949), 2C: 0.8% (4/511)). We consider nonessential genes which are rarely expressed as the negative control gene set since the differential dependencies are unlikely to be non-essential (unexpressed) for cellular activity.

### Analysis of CERES dependency probabilities

We also ran SuperDendrix on the dependency probabilities obtained from the Avana dataset. These probabilities are computed from reference gene sets of unexpressed genes and essential genes and attempt to quantify the probability that a CERES score represents a true dependency.

In the first module of SuperDendrix, we identified differential dependencies from the dependency probabilities using the following $3\sigma$ criterion that is similar to the $6\sigma$ criterion used in the previous analysis of RNAi screens.[19] First, we defined the direction of each gene dependency as increased if the majority of the cell lines have dependency probability less than 0.5 and decreased otherwise. Then we defined each gene dependency as a $3\sigma$ differential dependency if at least 20% of the cell lines – close to the average percentage (23%) of cell lines in the outlier components in two-component mixtures from CERES scores – have dependency probabilities more than three standard deviations away from the mean. Using the $3\sigma$ criterion, we selected 810 $3\sigma$ differential dependencies (804 increased and 6 decreased). 126 of the 810 $3\sigma$ differential dependencies are also 2C differential dependencies identified by SuperDendrix. The $3\sigma$ differential dependencies contain a significantly lower proportion of cancer genes from Cancer Gene Census (CGC) than 2C differential dependencies ($3\sigma$: 7.4%, 2C: 17.2%; $p \leq 1e-5$, two-sample proportion z-test).

SuperDendrix identified significant associations ($FDR \leq 0.2$) for 78 of the 810 $3\sigma$ differential dependencies. These include 15 differential dependencies with significant associations identified in the original analysis using 2C scores. The majority of the mutation sets for the 15 shared differential dependencies are similar: mutation sets for 10 differential dependencies are identical, and the mutation sets for 3 other differential dependencies contain at least one mutation in common. The 63 associations that were uniquely identified using dependency probabilities contain a significantly lower proportion of CGC cancer genes than the 112 associations that were identified uniquely using 2C scores (dependency probability: 4.8%, 2C score: 29.5%, p = 0.00005; two-sample proportion z-test).

### SuperDendrix analysis of all non-synonymous mutations

In the first module for classification and selection of OncoKB mutations, SuperDendrix identified 20,089 mutations that occur in OncoKB-annotated cancer genes from 328,667 non-synonymous mutations in the CCLE data and searched for associations using this subset.

For comparison, we also ran SuperDendrix without restricting to mutations annotated in OncoKB. Using all 328,667 non-synonymous mutations in 13,334 genes. SuperDendrix identified 121 differential dependencies with significant associations ($FDR \leq 0.2$), 80 of which are associated with sets containing multiple mutations. These include 77 differential dependencies that were identified from the analysis using OncoKB mutations only. The associations for the overlapping 77 differential dependencies from the two analyses are similar overall: 32 associated sets of mutations for 32 of these 77 differential dependencies are identical and another 33 mutation sets share at least one OncoKB mutation in common. Not surprisingly, while mutations identified in both analyses and mutations identified only in the OncoKB analysis are both significantly enriched ($p \leq 0.05$; hypergeometric test) for cancer genes from Cancer Gene Census (CGC), mutations in associations identified only when using all non-synonymous mutations are not enriched for CGC genes (p = 0.2, hypergeometric test). These associations require additional validation.

Thus, while OncoKB mutations account for only 6.1% of the non-synonymous mutations, they account for 62% of the differential dependencies with associations and 37.4% of the mutations found by SuperDendrix. This indicates that associations for differential dependencies are saturated by a small subset of mutations in known cancer genes selected by SuperDendrix

### Network analysis of pathway addiction

We performed an analysis that integrates the associations identified by SuperDendrix with prior knowledge of physical interactions in protein-protein interaction (PPI) networks. First, we add edges to the PPI network for each association between a mutation and a differential dependency identified by SuperDendrix. We then find subnetworks that are connected in physical interactions and dense in genetic dependencies. This approach automates some of the manual annotation that we performed to identify oncogenic pathway addiction.

We searched for the densest connected subnetworks of 6 different sizes (10, 15, 20, 25, 30, and 35 vertices) from a dual network of 176,839 physical interaction edges from HINT+HI network[114] - a combination of HINT and HI interaction networks - and 561 genetic dependency edges derived from SuperDendrix associations for 511 differential dependencies. We computed a $P$-value for each subnetwork using a permutation test by permuting genetic dependency edges as described in a previous study.[120] All of the densest

connected subnetworks we identified are statistically significant (p≤0.05). Note that the densest connected subnetwork of size 35 contains genes that span multiple addicted pathways including the NFE2L2 pathway, the MAPK pathway, and the Wnt pathway. Interestingly, this subnetwork also contains an association between TAZ dependency and the set, {TP53(*I*), TP53(O)} that is not statistically significant according to SuperDendrix (Figure S5). TAZ is a transcriptional regulator that has been identified as a key driver of various cancers.[121] The association of *TAZ* dependency with *TP53* mutations is consistent with a recent report[122] that mutant p53 leads to aberrant activation of the YAP/TAZ transcriptional regulator complex.

## Validation on the Sanger CRISPR-Cas9 screen data

We used the dataset from genome-wide CRISPR-Cas9 screens conducted as part of the Cancer Dependency Map at Wellcome Sanger Institute[20] to validate the associations identified by SuperDendrix from the Avana dataset of the Cancer Dependency Map at the Broad Institute.

First, we downloaded the dataset[20] [Release 1/4.5.2019] containing dependency scores computed from results of CRISPR screens across 324 cancer cell lines from the Project Score data portal and a list of mutations for the same cell lines from Cell Model Passports[113] data portal. We used quantile normalized log fold-changes as dependency scores and processed the mutation data using SuperDendrix OncoKB feature selection. We restricted our validation to 312 cell lines that contain at least one OncoKB mutation feature.

For each association identified by SuperDendrix in the Avana dataset, we compared the dependency scores of cell lines containing at least one of the features with dependency scores of the cell lines without any feature. We excluded the associations for which dependency or feature data is not available in the Score dataset. We found that associations between 45/110 differential dependencies and mutations and associations between 146/210 differential dependencies and cancer types and/or mutations identified by SuperDendrix are statistically significant in the Score dataset (p≤0.05; Wilcoxon rank sum test).

We find that many of the associations identified by SuperDendrix that did not validate in the Score dataset are in cancer types that were poorly represented in the Score dataset (Table S5).

## Comparison with other perturbation screen results

We compare the differential dependencies and mutation sets associated with these dependencies identified with our methods to the results of RNAi screening from Tsherniak et al.[19] and CRISPR-Cas9 screening from Behan et al.[20]

Tsherniak et al.[19] identified $6\sigma$ genes and associated genomic markers of these differential dependencies using RNAi screens of 501 cancer cell lines as part of the DepMap project. This analysis is distinct from ours in terms of the perturbation assay (RNAi instead of CRISPR) and score (DEMETER[19] instead of CERES), and in that Tsherniak et al. consider copy number aberrations and gene expression data – in addition to mutations – as potential genomic markers. There are 353 cell lines shared between the RNAi and CRISPR datasets.

We first compare in terms of differential dependencies. Tsherniak et al.[19] analyzed 6,305 profiles that pass quality control and identified 769 $6\sigma$ genes. 92 of these $6\sigma$ profiles are also among the 511 2C differential dependencies. Despite the small number of overlaps, the two sets of differential dependency profiles represent similar classes of proteins. In particular, both sets are significantly enriched for GO molecular functions such as *DNA binding* and *protein kinase activity*. They also contain similar proportion of CGC genes (Tsherniak et al. $6\sigma$: 12.1% p<0.01, 2C: 17.2% p<0.01). Genes that are unique to each set also capture similar GO molecular functions including nucleotide binding, protein binding, and G protein-coupled receptor activity and are both significantly enriched for CGC genes (p<0.01).

We next compare our results with Tsherniak et al.[19] in terms of biomarkers for differential dependency. Tsherniak et al. used a random forest-based approach to identify genomic features that are predictive of differential dependency, which they referred to as "marker dependency pairs" (MDPs). Using mutations, copy number aberrations, and gene expression, Tsherniak et al.[19] found MDPs for 426 of the 769 $6\sigma$ profiles in the RNAi data. However, only 10 of these correspond to mutation driven biomarkers. In contrast, SuperDendrix found significantly associated mutation sets in 127 of 511 2C differential dependencies in the CRISPR data. 7 biomarker associations (mutation driven) are identified by both methods. These include well-known associations such as oncogene addictions of *BRAF*, *NRAS*, and *KRAS*. Interestingly, associations identified only by SuperDendrix include those with strong evidence, such as *RAF1* dependency on *KRAS* or *NRAS* mutations, *STAG1* dependency on *STAG2* mutations, and *NFE2L2* dependency on *KEAP1* mutations.

As part of the DepMap project, Behan et al.[20] independently conducted genome-wide CRISPR-Cas9 loss-of-function screens in 324 cancer cell lines that include 178 cell lines from the Avana dataset. From a total of 18,009 knockout genes, they identified 628 priority targets based on combination of gene knockout effect across cell lines and their associations to biomarkers (single nucleotide variants, copy number variations, and microsatellite instability status). 148 of the priority targets are also among the 511 2C profiles from SuperDendrix. The two sets of genes are significantly enriched for GO molecular functions such as DNA binding, protein binding, and transcription regulator activity. They also contain similar proportion of CGC genes (priority targets:15.8% p≤0.01, 2C: 17.2% p≤0.01).

Behan et al. analyzed associations of gene knockout effects with genomic biomarkers within each cancer type using ANOVA. Associations that occur across multiple cancer types were aggregated and re-tested using a *t*-test across all cell lines. We compare our results to their associations to SNVs considering all cell lines since we do not test for cancer-type-specific biomarker

associations. Behan et al. identified a total of 77 significant biomarker associations (p ≤ 0.05) in 51 of the 628 priority target genes. However, only 16 associations for 14 genes are with SNV biomarkers. 3 of these (KRAS-KRASmut, PIK3CA-PIK3CAmut, GRB2-KRASmut) are also identified by SuperDendrix.

Overall, we are able to explain 127 of the 511 2C differential dependencies (24.9%) with mutations using SuperDendrix, 36 of which are associated with more than one mutation. In contrast, Tsherniak et al.[19] and Behan et al.[20] can each explain only 1.3% ($\frac{10}{769}$) and 2.5% ($\frac{16}{628}$) of their differential dependencies with mutation features. These findings indicate that our model, by searching for a set of approximately mutually exclusive mutations, has higher sensitivity for identifying associations between gene dependency biomarkers.

## Univariate analysis of the DepMap data

We find that the univariate analysis and SuperDendrix have some overlap in their reported associations, but also substantial differences (Figure S8). Only 65 differential dependencies are reported as associated with mutations by both methods (Figure S8A), while SuperDendrix and the univariate test report an additional 62 and 72 differential dependencies, respectively, to be associated with mutations (Figure S8B-C). The 62 differential dependencies reported uniquely by SuperDendrix contain a higher proportion of CGC cancer genes than those reported uniquely by the univariate analysis (12/62 for SuperDendrix versus 8/72 for univariate, p = 0.09; two-sample proportion test, Figure S8D). Moreover, the associations found uniquely by the univariate test are skewed toward associations involving the most frequently mutated genes and the cell lines with the most mutations in the dataset. In particular, the mutations in the associations reported uniquely by the univariate test have a higher average frequency than the mutations in associations reported uniquely by SuperDendrix (78.2 for univariate versus 41.6 for SuperDendrix, p = 0.011; t test, Figure S8E). Over a third (33/94) of the associations reported uniquely by the univariate test involve 3 frequent mutations, KRAS(A), BRAF(A), and TP53(*I*) that are mutated in 130, 65, and 495 cell lines, respectively. In contrast, because SuperDendrix examines combinations of mutations, it has higher sensitivity for finding associations with rare mutations. For example, SuperDendrix finds an association between *CCND3* dependency and *CCND3* activating mutation (5 cell lines), a previously reported oncogene addiction, as part of the mutation set {CCND3(A), LTB(C)}. Second, the difference in the number of associations reported uniquely by the univariate test and SuperDendrix is positively correlated (*R* = 0.39, p ≤ 2.2*e* − 16; Pearson correlation, Figure S8F) with the total number of mutations in the cell line. This suggests that the univariate method lacks specificity in cell lines with many mutations due to lack of a procedure to control for variable mutation rate of cell lines.

We conducted a systematic univariate analysis to search for associations between mutation features and differential dependencies. Specifically, for each mutation and each differential dependency we compare the CERES dependency scores in cell lines with and without the mutation using the Wilcoxon rank-sum test. We perform this test for all 897 mutations and 511 differential dependencies identified in the first module of SuperDendrix, for a total of 458,367 tests. This univariate analysis identified 201 significant associations (*FDR* ≤ 0.2) between 137 differential dependencies and 76 mutations (Figure S8), compared to 172 significant associations (*FDR* ≤ 0.2) between 127 differential dependencies and 84 mutations identified by SuperDendrix (Figure S8).

Next, we compared the associated mutations reported by SuperDendrix and the univariate test for the 65 differential dependencies that both methods reported to have associated mutations (Figure S8A). We found that for 35 of these 65 differential dependencies, both methods reported the same set of mutations. For the other 30 differential dependencies, the differences between methods were analogous to these described above for the differential dependencies unique to each method. In particular, the univariate method tended to report more associations with the most frequent mutations; e.g., BRAF(A) and KRAS(A). Examining the differential dependencies with the largest differences in the number of associated mutations also demonstrates a key difference between the univariate test and SuperDendrix. The differential dependency with the largest difference in the number of associated mutations is *WRN* (Figure S9); the univariate analysis reports 13 associated mutations while SuperDendrix reports only one of these: *KMT2B* inactivating mutation (Figure S9A-B). Importantly, KMT2B(*I*) is most strongly associated with *WRN* dependency among the 13 mutations found by the univariate test. Furthermore, the 12 additional mutations occur in 23 of the 24 cell lines that contain *KMT2B* mutation, indicating strong co-occurrence between these mutations (Figure S9C). Not surprisingly, the set of 13 mutations found by the univariate test have weaker SuperDendrix weight which scores mutual exclusivity of mutations and their association to differential dependency than the mutation reported by SuperDendrix (Figure S9D). This example illustrates one of the key differences between SuperDendrix and the univariate analysis: the univariate analysis evaluates each mutation association independently while SuperDendrix examines mutual exclusivity between mutations and thus avoids reporting overlapping, redundant associations.

Microsatellite instability (MSI) was previously reported to be associated with both *WRN* dependency and downregulation of *KMT2B*.[123] Therefore, we conducted an additional analysis of *WRN* dependency using the MSI status (available for 639 of 769 cell lines from the DepMap 20Q2 release) as an additional binary feature in the feature matrix of SuperDendrix. We used the MSI status for each cell line reported in Chan et al., 2019.[124] We find that *WRN* dependency is more significantly associated with KMT2B(*I*) mutation found by SuperDendrix than MSI status (KMT2B(*I*): 0.0000, MSI: 0.0611; *P*-value from SuperDendrix). We also confirmed that the strongest association with *WRN* dependency identified by SuperDendrix is KMT2B(*I*) when MSI is included in the feature matrix. Interestingly, while most (20/24) of the cell lines with KMT2B(*I*)mutation contain MSI, we find that KMT2B(*I*) is more specific to increased dependency on *WRN*; a higher fraction of the KMT2B(*I*) mutated cell lines are dependent on *WRN* than the MSI cell lines (KMT2B(*I*): 18/24, MSI: 22/41, Figure S10). It is possible that the higher significance and specificity of the association between *WRN* dependency and KMT2B(*I*) than MSI indicates that the methylation status of H3 histone mediated by the KMT2B(*I*) mutation may

represent a specific molecular mechanism in MSI status that confers the synthetic lethal interaction with *WRN*. Another alternative is that the MSI status of some cell lines is incorrect. Further validation studies will be necessary to distinguish the functional linkages between *WRN*, *KMT2B*, and MSI.

On the other hand, the univariate test misses interesting associations with rare mutations that are reported by SuperDendrix (Figure S11). For example, SuperDendrix reports a set of three mutations, {KEAP1(I), KEAP1(O), NFE2L2(A) } to be associated with increased dependency on *NFE2L2* (Figure S11A). In contrast, the univariate test reports only two of these mutations, KEAP1(I) and KEAP1(O). The association between NFE2L2(A) mutation and *NFE2L2* dependency is consistent with oncogene addiction and has been reported previously, but was missed by the univariate test because NFE2L2(A) is a rare mutation present in only 7/767 cell lines in the dataset (Figure S11B) Another interesting example is increased dependency on *FANCG*. SuperDendrix reports BRCA1(I), a relatively rare mutation occurring in 15/767 cell lines, to be associated with *FANCG* dependency (Figure S11C). Both FANCG and BRCA1 (also known as FANCS) are members of the FA-BRCA pathway that regulates DNA damage response and are novel candidates for synthetic lethal interaction. On the other hand, the univariate test reports an association between *FANCG* and the frequent but functionally unrelated mutation, BRAF(A) (65/767 cell lines) (Figure S11D). These examples again demonstrate the key difference between SuperDendrix and the univariate analysis: the univariate analysis evaluates each mutation association individually while SuperDendrix scores association between a set of mutually exclusive mutations enabling the identification of associations with rare mutations.

Taken together, these results show that the univariate test and SuperDendrix have different trade-offs in the identification of associations: the univariate test is confounded by mutation rate, reporting many associations with frequently mutated genes and in cell lines with high mutation rates. In contrast, SuperDendrix identifies associations with rarely mutated genes that are mutually exclusive of associations with more frequently mutated genes, but might miss some associations in samples with extremely high mutation rates (e.g., due to MSI) which lead to co-occurrence between driver and passenger mutations.

### Univariate analysis of cancer-type-specific differential dependencies

We conducted a systematic univariate analysis to search for associations between differential dependencies and combinations of cancer type and/or mutation features. We analyzed a total of 474,208 pairs consisting of one of 511 differential dependencies and one of 928 features (31 cancer types and 897 mutations). This univariate analysis identified 861 significant associations ($FDR \leq 0.2$) between 334 differential dependencies, 25 cancer types and 142 mutations (Figure S12), compared to 501 significant associations ($FDR \leq 0.2$) between 227 differential dependencies, 27 cancer types and 55 mutations identified by SuperDendrix (Figure S12).

We find a sizable difference between the associations identified by the univariate test and SuperDendrix. While 203 differential dependencies are reported by both methods to have associations (Figure S12A), the univariate test reports an additional 131 unique differential dependencies with associations, while SuperDendrix reports an additional 24 unique differential dependencies with associations (Figure S12B-C). We found that the associations reported uniquely by the univariate test are biased toward frequent features and cell lines with higher mutation rate, analogous to the results reported above with mutation features alone. Specifically, the features in associations reported uniquely by the univariate analysis have a higher average frequency than those in associations reported uniquely by SuperDendrix (univariate: 39.2, SuperDendrix: 26.7, p = 0.002; t test, Figure S12D). On the other hand, the features in associations reported uniquely by SuperDendrix that were not in associations reported by the univariate test are all rare features that occur in less than 20 cell lines (average frequency: 13.1, starred in Figure S12B)). In addition, the difference in the number of associations reported by the univariate test and SuperDendrix is positively correlated ($R$ = 0.5, p $\leq 2.2e - 16$; Pearson correlation, Figure S12E) with the total number of mutations in the cell line, indicating that some of the associations reported by the univariate test are likely false positives in cell lines with high numbers of mutations. These suggest two issues of the univariate test: The univariate test lacks sensitivity in features with low frequency and specificity in cell lines with many mutations because the univariate test evaluates each feature independently and lacks a procedure to control for variable mutation rate of cell lines. In contrast, SuperDendrix evaluates combinations of mutually exclusive features and controls for mutation rate of cell lines in the statistical test of its third module.

Next, we compared the features that were reported to be associated with the 203 differential dependencies identified by both methods (Figure S12A). We found that both methods reported the same sets of features for 52 of these 203 differential dependencies. Associations reported uniquely by the univariate test tended to include frequent features and cell lines with high mutation rates. Furthermore, the univariate test reported many differential dependencies to be associated with both a mutation and a cancer type where this mutation frequently occurred. For example, BRAF(A) is the mutation with most associations reported by the univariate test, and this occurs frequently in skin cancer (39/65 cell lines with BRAF(A) are skin cancer, fold-enrichment = 8.52, p = $1.2e - 39$; hypergeometric test, Figure S13A). Interestingly, 24 of the 34 differential dependencies reported by the univariate test to be associated with either BRAF(A)or skin cancer are reported as associated with both BRAF(A) and skin cancer (Figure S13B). On the other hand, none of the 26 differential dependencies reported by SuperDendrix to be associated with BRAF(A) or skin cancer are associated with both features. This again demonstrates the key difference between the univariate test and SuperDendrix that was described above: the univariate test evaluates each association independently and does not account for correlation between features while SuperDendrix examines mutual exclusivity of features and thus avoids redundant associations of correlated features. This difference is also apparent in the mutation with second most associations, KRAS(A). Cell lines with KRAS(A) mutation are significantly enriched for pancreatic cancer, colon cancer, lung cancer, and bile duct cancer (Figure S13C). 14 of the 34 differential dependencies reported by the univariate test to be

associated with KRAS(A) or these four enriched cancer types are associated with both KRAS(A) mutation and at least one of the enriched cancer types (Figure S13D). In contrast, SuperDendrix does not report any redundant associations in 40 differential dependencies associated with KRAS(A) or the enriched cancer types.

Taken together, these results indicate a similar tradeoff in the identification of associations described previously in the comparison of associations to mutations: While the univariate test reports a higher number of associations than SuperDendrix, its associations tend to include redundant associations between correlated features and are also biased toward cell lines with higher mutation rate. On the other hand, SuperDendrix prioritizes mutually exclusive features and selects the strongest associations, thus reporting fewer and less redundant associations.

## Comparison with UNCOVER

As noted in the introduction, there are two other methods to find associations between mutually exclusive mutations and gene perturbation scores: REVEALER[25] and UNCOVER.[28] REVEALER uses a greedy method to find mutually exclusive mutations associated with continuous phenotype. As noted previously,[28] the greedy method is slow and not scalable to the large-scale Avana dataset containing thousands of dependency profiles, and therefore was not compared with SuperDendrix. UNCOVER was developed concurrently with our development of SuperDendrix, and also solves a combinatorial optimization problem. However, there are several key differences between SuperDendrix and UNCOVER.

1. UNCOVER is applied directly to dependency scores, while SuperDendrix first identifies and scores differential dependencies using a mixture model.
2. UNCOVER combines all mutations in a gene into a single gene-level mutation, while SuperDendrix creates different mutation features (GENE(A), GENE(I), or GENE(O)) according to OncoKB annotations.
3. UNCOVER uses a different objective function in the optimization with positive and negative scores having asymmetric contribution to the objective.
4. UNCOVER lacks a model selection step and does not control for variability in the number of mutations across cell lines during its statistical test.

First, we highlight the difference between the SuperDendrix weight and the UNCOVER objective function which we reproduce below using the same notation from the SuperDendrix weight:

$$w(M) = \sum_{j \in \Gamma(M)} d_j - (c_j(M) - 1)p_j$$

This function consists of two terms, $d_j$ and $p_j$, that represent association to phenotype and penalty for co-occurring mutations. While UNCOVER uses the same linear term as SuperDendrix for biomarker-phenotype association, it uses a penalty term that has different values depending on the sign of the phenotype score. Specifically, if the phenotype score $d_j$ in cell line $j$ is positive, then UNCOVER sets the penalty $p_j$ to the *average* of the positive phenotype scores; alternatively, if the phenotype score $d_j$ in cell line $j$ is negative then UNCOVER sets the penalty $p_j$ to be the absolute value of the score.

Next, we compared SuperDendrix and UNCOVER on the same Avana dataset and found that the methodological differences between SuperDendrix and UNCOVER led to large qualitative and quantitative differences in results. For consistency with the original study,[28] we first standardized the CERES scores into z-scores and constructed gene-level mutation features by combining missense, nonsense, and frameshift mutations of each gene into a single feature. Then we ran UNCOVER using the standardized CERES scores of 2,074 $6\sigma$ profiles and 13,311 mutation features and 31 cancer-type features search for a set of 3 associated mutation features and 5 mutation and/or cancer-type features for both directions of dependency. UNCOVER reported 248 sets of mutations containing a total of 744 mutations with significant association ($p \leq 0.001$, Table S10), compared to 127 sets containing a total of 172 mutations for SuperDendrix. When the 31 cancer-type features were included, UNCOVER reported 860 sets of cancer-type features and/or mutations compared to 227 for SuperDendrix (Table S11).

There are multiple reasons for the larger number of associations predicted by UNCOVER. First, 138 of the 248 significant associations identified by UNCOVER are associations between dependencies that are not identified as differential dependencies by SuperDendrix. These include dependencies on *USP1* and *MAP3K2* whose dependency score distributions are unimodal (Figure S14A). Second, UNCOVER does not include a model selection procedure, and thus always returns mutation sets of the requested size (3 and 5 in these experiments). Of the 52 differential dependencies where both UNCOVER and SuperDendrix reported associated mutations in the same direction, UNCOVER's associated sets included 156 gene-level mutations (52 × 3), while SuperDendrix sets contain a total of 83 mutations (including GENE(A), GENE(I), and GENE(O) mutation features). 43 of the gene-level mutations identified by UNCOVER overlap the 83 mutations identified by SuperDendrix in the corresponding profile. The remaining 113 gene-level mutations found by UNCOVER are not included in SuperDendrix results. Notably, 63 of these 113 mutations contribute less than 20% to the corresponding weight of the mutation set. Across UNCOVER's 248 total significant associations, 18% of the significant mutations ($\frac{295}{744}$) contribute less than 20% to the set's weight. These mutations with small objective values are likely false positives. Finally, the permutation test used to evaluate statistical significance of UNCOVER's results does not control for variability in the number of mutations across cell lines. We found that the number of significant associations reported by UNCOVER in a cell line is

significantly correlated with the number of mutations in the cell line (Pearson correlation: $R = 0.66$ for mutations only; and $R = 0.63$ with cancer types included, Figure S14B-C), indicating that some of the associations reported by UNCOVER are likely false positives. In comparison, the correlation is much weaker in SuperDendrix results ($R = 0.36$ for mutations only; and $R = -0.01$ with cancer types included, Figure S14B-C)).

### Comparison with SELECT

The SELECT method[40] has three major differences from SuperDendrix. First, SELECT examines only correlations between mutations and does not compute associations between mutations and quantitative phenotypes. In contrast, SuperDendrix scores sets of mutations according to their association with a phenotype of interest. Second, SELECT scores pairs of mutations while SuperDendrix evaluates larger sets of mutations. Finally, SELECT combines all non-synonymous mutations in a gene into a single feature while SuperDendrix separates mutations in a gene into three features: "**A**ctivating," "**I**nactivating," and "**O**ther" according to OncoKB annotations.

Despite these differences, we used SELECT in a two-step procedure to identify associations between mutations and differential dependencies by first running SELECT on the mutation features derived by SuperDendrix and then applying the univariate test to identify associations between differential dependencies and the mutually exclusive mutations reported by SELECT. SELECT identified only 24 pairs of mutations (in a total of 33 genes) that are associated (via Wilcoxon rank-sum test) with 280 differential dependencies, compared to the 87 sets of mutations in 84 genes that are associated with 127 differential dependencies identified by SuperDendrix. Most of the SELECT associations are dominated by mutations in a small number of well-known cancer genes (Figure S15). For example, 46% of the differential dependencies reported by SELECT to have associated mutations are associated with frequent mutations: KRAS(A) (130 cell lines), BRAF(A) (65 cell lines), TP53(I) (495 mutations), or NRAS(A) (48 cell lines). In comparison, these four mutations are associated with only 27% of the differential dependencies reported by SuperDendrix. Overall, we found that the associations reported by SELECT are biased toward frequent mutations and cell lines with higher mutation rate. Specifically, the mutations in associations reported by SELECT have a higher average frequency than those in associations reported by SuperDendrix (SELECT: 64.8, SuperDendrix: 38.6, $p = 1.2e - 14$; t test, Figure S16A). In addition, the difference in the number of associations reported by SELECT and SuperDendrix is positively correlated ($R = 0.27$, $p = 2.02e - 14$; Pearson correlation, Figure S16B) with the total number of mutations in the cell line, indicating that some of the associations reported by SELECT are likely false positives in cell lines with high numbers of mutations. Lastly, SELECT does not find associations to single mutations or sets of three mutations as it only analyzes pairs of mutations. As a result, the majority (54/74) of the associations reported by SuperDendrix that include only a single mutation are not reported by SELECT. These include associations between *HRAS* dependency and *HRAS* mutation and between *PIK3CA* dependency and *PIK3CA* mutation which have been reported previously as oncogene addictions.

### Expression of lineage-specific transcription factors

SuperDendrix identified differential dependencies on 43 transcription factors that are significantly associated with specific cancer types. Cancer-type-specific dependencies on transcription factors have been reported previously to be associated also with expression of these genes.[19] Therefore, we compared the expression of the 43 transcription factors with their gene dependency to evaluate the importance of gene expression on lineage-specific gene dependency.

Our analysis revealed that expression of the dependent gene is strongly correlated with dependency on the majority of the transcription factors (Figure S6A). This indicates that expression of the dependent gene is important in addition to the lineage classification for predicting gene dependency. Interestingly, we find that elevated expression of the dependent gene is specific to the associated cancer types with strong dependency for many transcription factors. For example, most of the cell lines with increased dependency on *SOX10* and high *SOX10* expression correspond to Skin cancer (Figure S6B). The cancer-type-specificity of expression and dependency indicates that either expression of the dependent gene or the lineage classification is sufficient to predict *SOX10* dependency across cell lines. On the other hand, there are transcription factors where high gene expression is not specific to the associated cancer types. For example, increased dependency on *SOX9* is associated with 5 cancer types. Interestingly, many of the cell lines with high *SOX9* expression and increased dependency on *SOX9* are not part of the 5 associated cancer types (Figure S6C), indicating that lineage classification alone does not predict *SOX9* dependency in these cell lines. Furthermore, we find that many of the cell lines from Gastric cancer, one of the 5 cancer types associated with *SOX9* dependency, have low *SOX9* expression despite their increased dependency on the gene. Taken together, these findings demonstrate that both expression of the dependent gene and lineage classification are important for dependency on *SOX9* across cell lines.

### Decreased dependency on transcription factors

Two of the 43 transcription factors with cancer-type specific differential dependencies, *THAP1*, *TP53*, show *decreased dependency* in specific cancer types. For example, we find decreased dependency on *THAP1* in leukemia. THAP1 is known as a pro-apoptotic factor involved in regulating endothelial cell proliferation and linking PAWR to promyelocytic leukemia (PML) nuclear bodies (NB).[84,85] Interaction of PAWR and PML has been reported to trigger apoptosis.[86] Furthermore, PML is a tumor suppressor primarily expressed in blood vessels and a negative regulator of cell survival pathways.[84,86] These reports on lineage-specificity and function of THAP1 and PML suggest that knocking out *THAP1* which leads to loss of PML function resulted in decreased dependency or even prolonged cell survival in leukemia and lymphoma. We also find decreased dependency on *TP53* in BRAF(A), kidney, rhabdoid, and liposarcoma

cancer cell lines. A possible explanation for decreased dependency on *TP53* is its wild-type function as a tumor suppressor. A previous study reports that knocking out *TP53* in cells with functional wild-type *TP53* where p53 acts as a tumor suppressor will induce growth advantage in those cells [@giacomelli2018mutational]. In our results, we noticed that many of the rhabdoid and kidney cancer cell lines as well as skin cancer cell lines with BRAF(A) mutations contain wild-type *TP53*. We thus tested for association between decreased dependency on *TP53* and TP53(WT) as an additional feature using SuperDendrix. In fact, SuperDendrix identified a significant association between them (p ≤ 0.001), confirming that this is a decreased dependency conferred by inhibiting tumor suppressor activity of p53 in *TP53* wild-type cell lines as suggested previously.[87]

## ADDITIONAL RESOURCES

### Web browser for genetic dependency and mutation data

We release a public, open-source web browser to view and explore SuperDendrix results. Users can choose which genetic dependency profile and which mutations they want to view or preload an association identified as significant by the SuperDendrix software. The browser displays a waterfall plot, indicating the dependency score and mutation status in each cell line. It also includes two bar plots on top of the waterfall plot that indicate tissue type and number of mutations per cell line. Users can interact with the plots by scrolling over bars in the waterfall plot. On mouse over, the browser displays tooltips listing information about the given cell line such as tissue type. Users can also select a range of cell lines in the bar plot at the top to zoom in. The plots provide an easy way to quickly assess whether the dependency scores in cell lines with user-specified mutations or cancer types are extreme relative to the other cell lines. The code for the SuperDendrix browser is open-source at https://github.com/lrgr/superdendrix-explorer (Zenodo: https://doi.org/10.5281/zenodo.5878914), and the browser itself is publicly available at https://superdendrix-explorer.lrgr.io/.

