## [Document S2. Transparent peer review records for Park et al. · Cell Genomics]

SuperDendrix algorithm integrates genetic dependencies and genomic alterations across pathways and cancer types

Tae Yoon Park^{1,2,5}, Mark D.M. Leiserson^{3,5}, Gunnar W. Klau^{4,5}, Benjamin J. Raphael^{1,2,6,*}

Summary

Initial submission: Received : July 15th 2020

Scientific editor: Orli Bahcall

First round of review: Number of reviewers: 3
Revision invited : November 26th 2020
Revision received : July 16th 2021

Second round of review: Number of reviewers: 3
Accepted : January 24th 2022

Data freely available:

Code freely available:

This transparent peer review record is not systematically proofread, type-set, or edited. Special characters, formatting, and equations may fail to render properly. Standard procedural text within the editor's letters has been deleted for the sake of brevity, but all official correspondence specific to the manuscript has been preserved.

Referee reports, first round of review

Reviewer #1: The authors have developed an algorithm called SuperDendrix that can extract mutation-dependent essentialities from AVANA, a large scale CRISPR-based essentiality screen dataset, and also identify feature sets that are associated with these differential dependencies. They provide a rigorous statistical framework and overall the manuscript is well written, includes interesting case studies, the conclusions well supported and the work very interesting. The code is freely available and there is also an implementation in the form of a web server.

Of particular interest is the identification of reduced dependency genes upstream of a mutated gene vs increased downstream of one in pathways. This is not a required revision in my opinion, since the paper is a methods paper. However, it would be really cool to use the data as a whole and reconstruct per tissue or cancer type the addicted pathways in a data-driven way. This could point to new structure of the pathways in the specific conditions of that cancer type and that specific mutation and would be a very interesting finding in my opinion. Implementation of an automated way to extract addicted pathways from these kinds of datasets would be extremely interesting.

Reviewer #2: Here, Park et al tackle the important challenge of identifying associations between genetic dependencies and genomic alterations. Solving this challenge is fundamental for the success of cancer precision medicine.

Several previous approaches (referenced within the paper) have begun to explore this question focusing on existing data from large-scale cancer cell line screens, in several ways, either using earlier versions of the datasets (e.g. RNAi data), or by treating each predictive genomic alteration independently, or by modeling more complex interactions. However, the work is not complete since statistical power is limited and the search space is vast, making the identification of real associations from spurious ones challenging.

In this manuscript, the authors created a new computational approach, termed "SuperDendrix" and present a number of preliminary proof-of-concept observations. First, the authors use a novel 2-component (2C) mixture model to identify 492 (28%) genes from a larger set of 1,730 genes who show some evidence of outlier (6sigma) dependency, whereas perturbation causes cell lines to group into a major (labelled background) and minor (labelled responsive) sets of dependency scores. Second, the authors utilize somatic mutation interpretation information from OncoKB as well as lineage information to identify feature sets that trend towards mutual exclusivity and are associated with the minor 2C distribution. Finally, the authors calculate the statistical significance of each feature set-differential dependency association.

They make a series of interesting observations, including 32 relationships (6%) between somatic mutations and the 2C dependencies, including NFE2L2 pathway, RB and MAPK pathway observations and an additional 103 (21%) that relate to lineage including 41 strong transcription factor dependencies in each major lineage, including TCF3/CCND3 dependencies in blood cancers and lineage relationships with IGFR pathway dependencies.

Overall, this work is clearly presented, well reasoned, and sound computationally. While the discoveries are preliminary, they are solid and will be interesting to readers in that they present a new set of ways in looking at the existing data.

Minor concerns:

1. The DepMap data includes both normalized CERES scores as well as "dependency probabilities" (defined as: given a gene score, how likely to be a member of the non-essential distribution or the common essential distribution in that cell line). As noted in the DepMap portal, the dependency probability scores may be more suited for binary relationships for which cell lines are killed or not, which is the task the authors wish to use the Avana dataset for. Therefore, for the sake of rigor, it may be worth comparing results using CERES and dependency probability scores.
2. To focus their search, the authors limited the genomic alteration space explored to gene and mutation annotations found in OncoKB. This resulted in keeping only 363 genes, or 9,464 alterations out of 420,541 (2%) non-synonymous single-nucleotide mutations in the original data. It is unclear whether this bottleneck explains why the vast majority of the significant associations were well-established single gene associations and not sets of genomic alterations. Were the remaining 98% of the mutational data

discarded completely?

3. The main rationale behind SuperDendrix is the advantage over previous methods and the authors are to be commended on the excellent comparison to the UNCOVER method. However, given the extremely limited number of discoveries (~32 with mutations), it may be worth discussing whether this result supports or rejects the overall hypothesis that searching for sets of multiple mutational biomarkers is an endeavor that will reveal currently unknown biological relationships.

4. Lineage classification and the expression of lineage-specific transcription factors are closely related. The authors should determine whether expression of the dependent gene itself, the lineage classification, or both are important in Figure 5.

Reviewer #3: The authors present an original computational method that is able to identify sets of genomic alterations with a tendency to occur in a mutually exclusive fashion across a panel of immortalized human cancer cell lines and associate these with differential genetic dependency of such models. The authors build on their previous work and expertise in devising tools and methods for the identification of mutual exclusive (ME) genomic alterations going here one step forward and designing a tool that not only identifies such ME modules but also associate their presence with a continuous feature or read out. The range of applicability of SuperDendrix (the introduced method) is wide and general and here the authors show its usefulness by applying it to public available genomic perturbation data from the cancer dependency map project. While doing so the authors demonstrate that SuperDendrix is indeed able to associate combinations of genomics events to differential dependencies in cancer cells.

Briefly, this manuscript describes an elegant and rigorously described analytical method and tackle a timely and hard challenge. However the entity and potential impact of the presented results is quite modest. In the end the vast majority of associations unveiled by SuperDendrix encompasses well established oncogenetic addictions explicated by associations with increased dependency on a point mutated oncogene, which would be trivially unveiled with much simpler methods, even with a systematic univariate inference.

The second bunch of results are indeed richer and more interesting however, again the vast majority of hits are represented by individual cancer lineages and transcription factors dependencies, which would probably be unveiled via a simple differential dependency analysis contrasting cell lines from a given tissue to the rest of the panel.

In addition, in several other works many other computational approaches have been proposed to associate combination of genomic events (potentially ME events) to continuous read-outs, particularly drug responses across panel of cell lines.

In conclusion, without a proper follow up experimental validation or an in-silico validation on an independent dataset (as suggested below), and a more rigorous comparison with other existing methods, the current content of this submission is not sounding enough to warrant publication on Cell Genomics and it might be more appropriate for a more specific bioinformatics journal.

The author should have more convincingly shown advantages of SuperDenrix over other much simpler analytical methods.

Other points:

* The authors should also compare the outcomes of their strategy to identify differential essential genes with the method based on the normLTR score defined in PMID: 28753431

* While mentioning approaches that attempt to identify associations between differential cancer-dependencies/drug-responses and combination of genomic events, thus sets of multiple biomarkers, the approach presented in PMID: 32437684 could be cited and briefly discussed. In addition, the authors should note that many other previously published works described random-forests and penalized

regression models associating groups of genomics features to differential cancer dependencies detected using drug response as readouts. The author should mention this, potentially citing some of these works: PMID: 22460905, PMID: 27397505, PMID: 23180760, PMID: 26482930, PMID: 23993102.

* How Superdendrix perform when applied to drug response data instead of gene essentiality scores? an interesting additional study could asses the extent of agreement between dendrix outcomes across gene-essentiality/drug-response, which might serve as the basis to elucidate drug MoA (as explored in PMID: 3262796)

* The author should compare their algorithm and strategy with those presented in PMID: 28756993 or discuss commonalities/differences

* The authors claim that they have implemented an interactive tool for viusalising/exploring their results, however the link to superdendrix-explorer is broken and I haven't been able to assess this.

* The Broad DepMap dataset used in this manuscript is quite outdated. A 20Q2 version is now available. The authors might consider revamping their analysis with this more recent version. Even better the authors might use the Sanger dependency map CRISPR-screens data (available at <https://score.depmap.sanger.ac.uk/>) to independently validate the superdendrix associations. Following this, they could reperform their analysis on a recent joint dataset of cancer dependencies integrating both Sanger and Broad CRISPR screens (available at <https://depmap.org/broad-sanger/>) encompassing data for > 700 cell lines, for which precomputed/batch-corrected CERES scores are available.

minor points:

* Figure legends should be self-explicative, it is not clear without reading the main text what the 2C in figure 1B legend refers to

* I would change the color scheme associated with activating/inactivating mutations in fig1a. This is misleading as it seems to match the background/outlier distributions' distinction.

Authors' response to the first round of review

We thank the reviewers for their constructive comments on our submitted manuscript. We provide a point-by-point response to all comments below, and have revised our manuscript to address these comments. Reviewer comments are listed in black text with our responses in blue text. In the revised manuscript, we note the substantially revised parts in blue text.

First, we note that in response to reviewer comments, we updated our analysis from the older 19Q1 Broad DepMap release to the newer 20Q2 release. This newer release contains results of CRISPR screens of 18,119 genes across 769 cell lines and includes a total of 20,089 mutations in 897 OncoKB mutation features of 621 genes. In this new dataset, SuperDendrix identifies 127 differential dependencies compared to 32 in the previous analysis that are significantly associated ($FDR \leq 0.2$) with mutations including 36 sets of multiple mutations compared to 9 in the previous analysis).

The new analysis includes all of the associations that were highlighted in the main text of the original manuscript -- including oncogenic pathway additions in the NFE2L2 pathway, RB1 pathway and MAPK pathway as well as cancer-type-specific pathway additions in the TCF3 pathway and IGF1R pathway. We updated all of the figures in the manuscript with this new analysis.

Reviewer #1: The authors have developed an algorithm called SuperDendrix that can extract mutation-

dependent essentialities from AVANA, a large scale CRISPR-based essentiality screen dataset, and also identify feature sets that are associated with these differential dependencies. They provide a rigorous statistical framework and overall the manuscript is well written, includes interesting case studies, the conclusions well supported and the work very interesting. The code is freely available and there is also an implementation in the form of a web server.

Of particular interest is the identification of reduced dependency genes upstream of a mutated gene vs increased downstream of one in pathways. This is not a required revision in my opinion, since the paper is a methods paper. However, it would be really cool to use the data as a whole and reconstruct per tissue or cancer type the addicted pathways in a data-driven way. This could point to new structure of the pathways in the specific conditions of that cancer type and that specific mutation and would be a very interesting finding in my opinion. Implementation of an automated way to extract addicted pathways from these kinds of datasets would be extremely interesting.

We thank the reviewer for the constructive comment on data-driven analysis of results from SuperDendrix analysis. We addressed this comment in the following two ways.

First, we annotated each oncogenic pathway addiction identified from SuperDendrix results by cancer type, and recorded these in a new Table R1 below. We found that some pathway addictions were present in many cancer types (e.g. MAPK and IGF1R) while others were specific to a few related cancers, e.g. blood-cancer-specific addiction to the TCF3 pathway.

Pathway	Cancer types (Number of Associated/Total cell lines)
NFE2L2	Esophageal (3/25), Liver (3/22), Lung (17/106)
RB1	Bladder (3/29), Brain (6/60), Lung (9/106)
MAPK	Bile duct (12/26), Bladder (5/29), Brain (5/60), Breast (5/34), Colon (19/36), Endometrial (3/26), Esophageal (3/26), Gastric (7/26), Leukemia (12/41), Liver (3/22), Lung (30/106), Myeloma (10/20), Ovarian (8/42), Pancreatic (29/34), Sarcoma (4/12), Skin (47/54), Thyroid (3/6)
TCF3	Leukemia (16/41), Lymphoma (13/21), Myeloma (20/20)
IGF1R	Bile Duct (15/26), Bladder (7/29), Brain (3/60), Breast (9/34), Colon (9/36), Endometrial (4/26), Esophageal (3/25), Ewing's Sarcoma (15/15), Gastric (4/26), Head and Neck (4/33), Leukemia (17/41), Lung (3/106), Myeloma (19/20), Neuroblastoma (17/19), Ovarian (6/42), Pancreatic (23/34), Rhabdomyosarcoma (10/11)
WNT	Bile Duct (4/26), Colon (29/36), Gastric (7/26), Lung (5/106)

Table R1: Cancer types of pathway addiction. The cancer types reported by SuperDendrix to have associations between mutations in a pathway and a differential dependency in the same pathway.

Dependent cell lines are those with an increased dependency in the reported pathway. Cancer types with fewer than 3 associated cell lines are not shown.

Second, we agree that extending our method to automatically identify pathway addiction in a data-driven way is an interesting future direction. As a preliminary step in this direction, we performed a new analysis integrating the associations identified by SuperDendrix with prior knowledge of physical interactions in protein-protein interaction (PPI) networks. In brief, we add edges to the PPI network for each association between a mutation and a differential dependency identified by SuperDendrix. We then find subnetworks that are connected in physical interactions and dense in genetic dependencies. This approach automates some of the manual annotation that we performed to identify oncogenic pathway addiction. Using this approach, we identified a subnetwork containing genes in multiple addicted pathways including the NFE2L2, MAPK, and WNT pathways. We describe this analysis in a new section of the revised manuscript "*Network analysis of pathway addiction*", which we reproduce below.

"We performed an analysis that integrates the associations identified by SuperDendrix with prior knowledge of physical interactions in protein-protein interaction (PPI) networks. First, we add edges to the PPI network for each association between a mutation and a differential dependency identified by SuperDendrix. We then find subnetworks that are connected in physical interactions and dense in genetic dependencies. This approach automates some of the manual annotation that we performed to identify oncogenic pathway addiction.

We searched for the densest connected subnetworks of 6 different sizes ($K=[10, 15, 20, 25, 30, 35]$) from a dual network of 176,839 physical interaction edges from HINT+HI network - a combination of HINT and HI interaction networks - and 561 genetic dependency edges derived

from SuperDendrix associations for 511 differential dependencies. We computed a P -value for each subnetwork using a permutation test by permuting genetic dependency edges as described in PMID: 30423088. All of the densest connected subnetworks we identified are statistically significant ($P \leq 0.05$). Note that the densest connected subnetwork of size 35 contains genes that span multiple addicted pathways including the NFE2L2 pathway, the MAPK pathway, and the Wnt pathway. Interestingly, this subnetwork also contains an association between TAZ dependency and the set, {TP53(I), TP53(O)} that is not statistically significant according to SuperDendrix (Figure S5). TAZ is a transcriptional regulator that has been identified as a key driver of various cancers (PMID: 27300434). The association of TAZ dependency with TP53 mutations is consistent with a recent report (PMID: 28166194) that mutant p53 leads to aberrant activation of the YAP/TAZ transcriptional regulator complex."

Figure S5 from the revised manuscript: The densest connected subnetwork of 35 nodes in a dual network containing physical interactions from the HINT+HI network and genetic interactions from associations reported by SuperDendrix. The subnetwork contains genes across multiple pathways including the NFE2L2 pathway, the MAPK pathway,

and the Wnt pathway as well as a novel association between *TAZ* dependency and mutations in *TP53* that is not reported in the SuperDendrix analysis. The nodes represent differential dependencies or genes with mutations. Black edges correspond to physical interactions in the PPI network, and red edges correspond to associations between gene dependency and mutation identified by SuperDendrix.

Finally, we added the following sentence to the Discussion section to indicate future work in this direction.

“... ”

Fourth, further automating the identification of cancer-type specific oncogenic pathway addiction by integrating genetic dependencies identified by SuperDendrix with prior knowledge of biological pathways could aid in the discovery of non-canonical cancer vulnerabilities associated with rare mutations in the oncogenic pathways.

... ”

Reviewer #2: Here, Park et al tackle the important challenge of identifying associations between genetic dependencies and genomic alterations. Solving this challenge is fundamental for the success of cancer precision medicine.

Several previous approaches (referenced within the paper) have begun to explore this question focusing on existing data from large-scale cancer cell line screens, in several ways, either using earlier versions of the datasets (e.g. RNAi data), or by treating each predictive genomic alteration independently, or by modeling more complex interactions. However, the work is not complete since statistical power is limited and the search space is vast, making the identification of real associations from spurious ones challenging.

In this manuscript, the authors created a new computational approach, termed "SuperDendrix" and present a number of preliminary proof-of-concept observations. First, the authors use a novel 2-component (2C) mixture model to identify 492 (28%) genes from a larger set of 1,730 genes who show some evidence of outlier (6sigma) dependency, whereas perturbation causes cell lines to group into a major (labelled background) and minor (labelled responsive) sets of dependency scores. Second, the authors utilize somatic mutation interpretation information from OncoKB as well as lineage information to identify feature sets that trend towards mutual exclusivity and are associated with the minor 2C distribution. Finally, the authors calculate the statistical significance of each feature set-differential dependency association.

They make a series of interesting observations, including 32 relationships (6%) between somatic mutations and the 2C dependencies, including NFE2L2 pathway, RB and MAPK pathway observations and an additional 103 (21%) that relate to lineage including 41 strong transcription factor dependencies in each major lineage, including TCF3/CCND3 dependencies in blood cancers and lineage relationships with IGFR pathway dependencies.

Overall, this work is clearly presented, well reasoned, and sound computationally. While the discoveries are preliminary, they are solid and will be interesting to readers in that they present a new set of ways in looking at the existing data.

We thank the reviewer for their positive comments on the manuscript.

Minor concerns:

1. The DepMap data includes both normalized CERES scores as well as "dependency probabilities" (defined as: given a gene score, how likely to be a member of the non-essential distribution or the common essential distribution in that cell line). As noted in the DepMap portal, the dependency probability scores may be more suited for binary relationships for which cell lines are killed or not, which is the task the authors wish to use the Avana dataset for. Therefore, for the sake of rigor, it may be worth comparing results using CERES and dependency probability scores.

We ran SuperDendrix using the dependency probability scores, as suggested. In brief, we found 78 differential dependencies with significant associations compared to 127 found by SuperDendrix, with 15 in common. We suspect that many of these additional discoveries are false positives as we describe in a new section of the revised manuscript "*Analysis of CERES dependency probabilities*" reproduced below.

"We also ran SuperDendrix on the dependency probabilities obtained from the Avana dataset. These probabilities are computed from reference gene sets of unexpressed genes and essential genes, and attempt to quantify the probability that a CERES score represents a true dependency.

In the first module of SuperDendrix, we identified differential dependencies from the dependency probabilities using the following 3σ criterion that is similar to the 6σ criterion used in PMID: 28753430. First, we defined the direction of each gene dependency as *increased* if the majority of the cell lines have dependency probability less than 0.5 and *decreased* otherwise. Then we defined each gene dependency as a 3σ differential dependency if at least 20% of the cell lines – close to the average percentage (23%) of cell lines in the outlier components in two-component mixtures from CERES scores – have dependency probabilities more than three standard deviations away from the mean. Using the 3σ criterion, we selected 810 3σ differential dependencies (804 increased and 6 decreased). 126 of the 810 3σ differential dependencies are also 2C differential dependencies identified by SuperDendrix. The 3σ differential dependencies contain a significantly lower proportion of cancer genes from Cancer Gene Census (CGC) than 2C differential dependencies (3σ : 7.4%, 2C: 17.2%; P-value $\leq 1e-5$, two-sample proportion z-test).

SuperDendrix identified significant associations (FDR ≤ 0.2) for 78 of the 810 3σ differential dependencies. These include 15 differential dependencies with significant associations identified in the original analysis using 2C scores. The majority of the mutation sets for the 15

shared differential dependencies are similar: mutation sets for 10 differential dependencies are identical, and the mutation sets for 3 other differential dependencies contain at least one mutation in common. The 63 differential dependencies reported uniquely using dependency probabilities contain a significantly lower proportion of CGC cancer genes than the 112 differential dependencies reported uniquely using 2C scores (dependency probability: 4.8%, 2C score: 29.5%, P-value = 0.00005; two-sample proportion z-test)."

2. To focus their search, the authors limited the genomic alteration space explored to gene and mutation annotations found in OncoKB. This resulted in keeping only 363 genes, or 9,464 alterations out of 420,541 (2%) non-synonymous single-nucleotide mutations in the original data. It is unclear whether this bottleneck explains why the vast majority of the significant associations were well-established single gene associations and not sets of genomic alterations. Were the remaining 98% of the mutational data discarded completely?

Yes, in the previous analysis we focused our analysis on the 2% of mutations that were annotated in OncoKB. However, following the reviewer's suggestion, we ran SuperDendrix using *all* non-synonymous mutations. Note that in response to a comment from R3, we updated all analyses to the 20Q2 Broad DepMap (Avana) dataset which contains 328,667 non-synonymous single-nucleotide mutations from 767 cell lines. Interestingly, we found that SuperDendrix identified similar associations using either the full set of mutations or restricted to the mutations annotated in OncoKB. We added this analysis to a new section "*SuperDendrix analysis of all non-synonymous mutations*" of the revised manuscript which we copy here for reference.

"In the first module for classification and selection of OncoKB mutations, SuperDendrix identified 20,089 mutations that occur in OncoKB-annotated cancer genes from 328,667 non-synonymous mutations in the CCLE data and searched for associations using this subset.

For comparison, we also ran SuperDendrix without restricting to mutations annotated in OncoKB. Using all 328,667 non-synonymous mutations in 13,334 genes. SuperDendrix identified 121 differential dependencies with significant associations ($FDR \leq 0.2$), 80 of which are associated with sets containing multiple mutations. These include 77 differential dependencies that were identified from the analysis using OncoKB mutations only. The associations for the overlapping 77 differential dependencies from the two analyses are similar overall: 32 associated sets of mutations for 32 of these 77 differential dependencies are identical and another 33 mutation sets share at least one OncoKB mutation in common. Not surprisingly, while mutations identified in both analyses or mutations identified in only the OncoKB analysis are both significantly enriched ($P\text{-value} \leq 0.05$; hypergeometric test) for cancer genes from Cancer Gene Census (CGC), mutations in associations identified only when using all non-synonymous mutations are not enriched for CGC genes ($P\text{-value} = 0.2$, hypergeometric test). These associations require additional validation.

Thus, while OncoKB mutations account for only 6.1% of the nonsynonymous mutations, they account for 62% of the differential dependencies with associations and 37.4% of the mutations found by SuperDendrix. This indicates that associations for differential dependencies are saturated by a small subset of mutations in known cancer genes selected by SuperDendrix.”

3. The main rationale behind SuperDendrix is the advantage over previous methods and the authors are to be commended on the excellent comparison to the UNCOVER method. However, given the extremely limited number of discoveries (~32 with mutations), it may be worth discussing whether this result supports or rejects the overall hypothesis that searching for sets of multiple mutational biomarkers is an endeavor that will reveal currently unknown biological relationships.

First, we note that SuperDendrix identified a much larger number of discoveries in our updated analysis with the more recent 20Q2 DepMap data (127 vs. 32 previously); these include 36 of associations between sets of multiple mutations (compared to 9 previously).

To further address the reviewer’s question about the benefits of examining multiple mutational biomarkers, we conducted a systematic univariate analysis of the 20Q2 DepMap data to search for associations between single mutation features. In brief, we find that SuperDendrix identifies many associations not discovered by univariate analysis and is more robust against confounding due to mutation rate. We describe this analysis in “*Univariate analysis of the DepMap data*” section of the revised manuscript which is reproduced in the response to comment #1 in R3 comments.

4. Lineage classification and the expression of lineage-specific transcription factors are closely related. The authors should determine whether expression of the dependent gene itself, the lineage classification, or both are important in Figure 5.

We addressed this point by analyzing the correlation between gene expression and dependency scores of lineage-specific transcription factors. In brief, we found that both the expression of the dependent gene and the lineage classification are important for cancer-type-specific gene dependency. We describe this analysis in the section “*Expression of lineage-specific transcription factors*” of the revised manuscript, which we reproduce below.

“SuperDendrix identified differential dependencies on 43 transcription factors that are significantly associated with specific cancer types. Cancer-type-specific dependencies on transcription factors have been reported previously to be associated also with expression of these genes (PMID: 28753430). Therefore, we compared the expression of the 43 transcription factors with their gene dependency to evaluate the importance of gene expression on lineage-specific gene dependency.

Our analysis revealed that expression of the dependent gene is strongly correlated with dependency for the majority of the transcription factors (Figure S6A). This indicates that

expression of the dependent gene is important in addition to the lineage classification for predicting gene dependency. Interestingly, we find that elevated expression of the dependent gene is specific to the associated cancer types with strong dependency for many transcription factors. For example, most of the cell lines with increased dependency on SOX10 and high SOX10 expression correspond to Skin cancer (Figure S6B). The cancer-type-specificity of expression and dependency indicates that either expression of the dependent gene or the lineage classification is sufficient to predict SOX10 dependency across cell lines. On the other hand, there are transcription factors where high gene expression is not specific to the associated cancer types. For example, increased dependency on SOX9 is associated with 5 cancer types. Interestingly, many of the cell lines with high SOX9 expression and increased dependency on SOX9 are not part of the 5 associated cancer types (Figure S6C), indicating that lineage classification alone does not predict SOX9 dependency in these cell lines. Furthermore, we find that many of the cell lines from Gastric cancer, one of the 5 cancer types associated with SOX9 dependency, have low SOX9 expression despite their increased dependency on the gene. Taken together, these findings demonstrate that both expression of the dependent gene and lineage classification are important for dependency on SOX9 across cell lines.”

Figure S6 from the revised manuscript: Dependency on lineage-specific transcription factors is associated with both expression of the dependent gene and lineage classification. (A) Boxplot of Pearson correlation coefficients between expression of the dependent gene and its dependency scores across all cell lines. (B) SOX10 expression vs. CERES dependency scores of SOX10. The majority of the cell lines with increased dependency on SOX10 have elevated SOX10 expression and are Skin cancer cell lines. (C) SOX9 expression vs. CERES dependency scores of SOX9. Many of the dependent cell lines with high SOX9 expression are from other cancer types. Also, Gastric cancer cell lines with increased dependency on SOX9 have low SOX9 expression.

Reviewer #3: The authors present an original computational method that is able to identify sets of genomic alterations with a tendency to occur in a mutually exclusive fashion across a panel of immortalized human cancer cell lines and associate these with differential genetic dependency of such models. The authors build on their previous work and expertise in devising tools and methods for the identification of mutual exclusive (ME) genomic alterations going here one step forward and designing a tool that not only identifies such ME modules but also associate their presence with a continuous feature or read out. The range of applicability of SuperDendrix (the introduced method) is wide and general and here the authors show its usefulness by applying it to public available genomic perturbation data from the cancer dependency map project. While doing so the authors demonstrate that SuperDendrix is indeed able to associate combinations of genomics events to differential dependencies in cancer cells.

Briefly, this manuscript describes an elegant and rigorously described analytical method and tackle a timely and hard challenge. However the entity and potential impact of the presented results is quite modest. In the end the vast majority of associations unveiled by SuperDendrix encompasses well established oncogenetic addictions explicated by associations with increased dependency on a point mutated oncogene, which would be trivially unveiled with much simpler methods, even with a systematic univariate inference.

We thank the reviewer for the positive comments about the SuperDendrix method. Regarding the comments that the results are “quite modest”, we revised our analysis in response to the comments of all reviewers which resulted in an expansion of our results in two major directions.

- 1) We applied SuperDendrix to a more recent version of the DepMap data (20Q2) resulting in a much larger number of differential dependencies with statistically significant associations to mutations (127 in updated analysis vs. 32 previously). These new results include 36 differential dependencies that are associated with sets of multiple, mutually exclusive mutations (vs 9 previously).
- 2) We compared the SuperDendrix results to a univariate analysis of associations between the 511 differential dependencies and 897 mutation features in the 20Q2 data. We found that while the univariate analysis identified more associations than SuperDendrix (201 for univariate vs. 172 for SuperDendrix), many of the associations found only by the univariate analysis involve frequent mutations, highly mutated cell lines, and/or co-occur with other more meaningful mutations. On the other hand, SuperDendrix is more robust against these confounders and identifies associations that are more specific to cancer genes including the associations to rare mutations that are **not** detected in the univariate analysis. We describe this analysis in a new section, “*Univariate analysis of the DepMap data*” of the revised manuscript which we reproduce below.

“We conducted a systematic univariate analysis to search for associations between mutation features and differential dependencies. Specifically, for each mutation and each differential dependency we compare the CERES dependency scores in cell lines with and without the mutation using the Wilcoxon rank-sum test. We perform this test for all 897 mutations and 511 differential dependencies identified in the first module of SuperDendrix, for a total of 458,367 tests. This univariate analysis identified 201 significant associations ($FDR \leq 0.2$) between 137

differential dependencies and 76 mutations (Figure S8), compared to 172 significant associations ($FDR \leq 0.2$) between 127 differential dependencies and 84 mutations identified by SuperDendrix (Figure S8).

We find that the univariate analysis and SuperDendrix have some overlap in their reported associations, but also substantial differences (Figure S8). Only 65 differential dependencies are reported as associated with mutations by both methods (Figure S8A), while SuperDendrix and the univariate test report an additional 62 and 72 differential dependencies, respectively, to be associated with mutations (Figure S8B-C). The 62 differential dependencies reported uniquely by SuperDendrix contain a higher proportion of CGC cancer genes than those reported uniquely by the univariate analysis (12/62 for SuperDendrix vs 8/72 for univariate, P -value=0.09;

two-sample proportion test, Figure S8D). Moreover, the associations found uniquely by the univariate test are skewed toward associations involving the most frequently mutated genes and the cell lines with the most mutations in the dataset. In particular, the mutations in the associations reported uniquely by the univariate test have a higher average frequency than the mutations in associations reported uniquely by SuperDendrix (78.2 for univariate vs. 41.6 for SuperDendrix, P -value = 0.011; t -test, Figure S8E). Over a third (33/94) of the associations reported uniquely by the univariate test involve 3 frequent mutations, KRAS(A), BRAF(A), and TP53(I) that are mutated in 130, 65, and 495 cell lines, respectively. In contrast, because SuperDendrix examines combinations of mutations, it has higher sensitivity for finding associations with rare mutations. For example, SuperDendrix finds an association between CCND3 dependency and CCND3 activating mutation (5 cell lines), a previously reported oncogene addiction, as part of the mutation set {CCND3(A), LTB(O)}. Second, the difference in the number of associations reported uniquely by the univariate test and SuperDendrix is positively correlated ($R = 0.39$, P -value $< 2.2e-16$; Pearson correlation, Figure S8F) with the total number of mutations in the cell line. This suggests that the univariate method lacks specificity in cell lines with many mutations due to lack of a procedure to control for variable mutation rate of cell lines.

Next, we compared the associated mutations reported by SuperDendrix and the univariate test for the 65 differential dependencies that both methods reported to have associated mutations (Figure S8A). We found that for 35 of these 65 differential dependencies, both methods reported the same set of mutations. For the other 30 differential dependencies, the differences between methods were analogous to those described above for the differential dependencies unique to each method. In particular, the univariate method tended to report more associations with the most frequent mutations; e.g. BRAF(A) and KRAS(A). Examining the differential dependencies with the largest differences in the number of associated mutations also demonstrates a key difference between the univariate test and SuperDendrix. The differential dependency with the largest difference in the number of associated mutations is WRN (Figure S9); the univariate analysis reports 13 associated mutations while SuperDendrix reports only one of these: KMT2B inactivating mutation (Figure S9A-B). Importantly, KMT2B is most strongly associated with WRN dependency among the 13 mutations found by the univariate test. Furthermore, the 12 additional mutations occur in 23 of the 24 cell lines that contain KMT2B mutation, indicating strong co-occurrence between these mutations (Figure S9C). Not surprisingly, the set of 13

mutations found by the univariate test have weaker SuperDendrix weight which scores mutual exclusivity of mutations and their association to differential dependency than the mutation reported by SuperDendrix (Figure S9D). The association between WRN dependency and KMT2B inactivating mutation is consistent with previous reports that microsatellite instability (MSI) status of cell lines is associated with WRN dependency and downregulation of KMT2B (PMID: 31978347). This example illustrates one of the key differences between SuperDendrix and the univariate analysis: the univariate analysis evaluates each mutation association *independently* while SuperDendrix examines mutual exclusivity between mutations and thus avoids reporting overlapping, redundant associations.

On the other hand, the univariate test misses interesting associations with rare mutations that are reported by SuperDendrix (Figure S10). For example, SuperDendrix reports a set of three mutations, {KEAP1(I), KEAP1(O), NFE2L2(A)} to be associated with increased dependency on NFE2L2 (Figure S10A). In contrast, the univariate test reports only two of these mutations, KEAP1(I) and KEAP1(O). The association between NFE2L2(A) mutation and NFE2L2 dependency is consistent with oncogene addition and has been reported previously, but was missed by the univariate test because NFE2L2(A) is a rare mutation present in only 7/767 cell lines in the dataset (Figure S10B) Another interesting example is increased dependency on FANCG. SuperDendrix reports BRCA1(I), a relatively rare mutation occurring in 15/767 cell lines, to be associated with FANCG (Figure S10C). Both FANCG and BRCA1 (also known as FANCS) are members of the FA-BRCA pathway that regulates DNA damage response and are novel candidates for synthetic lethal interaction. On the other hand, the univariate test reports an association between FANCG and the frequent but functionally unrelated mutation, BRAF(A) (65/767 cell lines) (Figure S10D). These examples again demonstrate the key difference between SuperDendrix and the univariate analysis: the univariate analysis evaluates each mutation association *individually* while SuperDendrix scores association between a *set* of mutually exclusive mutations enabling the identification of associations with rare mutations.

Taken together, these results show that the univariate test and SuperDendrix have different tradeoffs in the identification of associations: the univariate test is confounded by mutation rate, reporting many associations with frequently mutated genes and in cell lines with high mutation rates. In contrast, SuperDendrix identifies associations with rarely mutated genes that are mutually exclusive of associations with more frequently mutated genes, but might miss some associations in samples with extremely high mutation rates (e.g. due to MSI) which lead to co-occurrence between driver and passenger mutations.

Figure S8 from the revised manuscript: Associations between differential dependencies (rows) and mutations (columns) identified by SuperDendrix and the univariate test. (A) Differential dependencies that are reported by both methods to have associated mutations. (B) Associations reported uniquely by SuperDendrix. (C) Associations reported uniquely by the univariate test. (D-F) Associations identified by the univariate test contain fewer cancer genes and are biased towards frequent mutations and cell lines with a high number of mutations. (D) Percentage of differential dependencies reported uniquely by SuperDendrix or uniquely by the univariate test that are in the Cancer Gene Census (CGC)

(E) Mutations in associations identified uniquely by the univariate test have higher frequency than those identified uniquely by SuperDendrix (univariate: 78.2, SuperDendrix: 41.6, P-value = 0.011; t-test). Mutation TP53(I) with outlier frequency (495 cell lines) was excluded from comparison. (F) The difference between the number of associations reported for a cell line by the univariate test and by SuperDendrix is positively correlated with the number of mutations in the cell line (R=0.39, P-value < 2.2e-16; Pearson correlation).

Figure S9 from the revised manuscript: Comparison of mutations reported by SuperDendrix and univariate test to be associated with increased dependency on WRN.

(A) Waterfall plot of WRN dependency scores colored according to presence of the 13 mutations reported by univariate test to be associated with WRN dependency. Two mutations, AXIN2(I) and XRCC2(I), that are not exclusive to any cell line are included in the “Multiple mutations” category. (B) Waterfall plot for the single mutation KMT2B(I) reported by SuperDendrix to be associated with WRN dependency shows that most of the cell lines with largest WRN dependency have KMT2B(I) mutations. (C) Mutation matrix for 13 mutations reported by univariate test and SuperDendrix to be associated with WRN dependency shows extensive co-occurrence between mutations. (D) SuperDendrix weight of KMT2B(I) mutation and set of 13 mutations.

Fig. S10 from the revised manuscript: Univariate test misses associations with rare mutations.

(A) Waterfall plot of NFE2L2 dependency scores colored according to presence of the 3 mutations reported by SuperDendrix to be associated with NFE2L2 dependency. The univariate test misses the association with a rare mutation, NFE2L2(A) (7 cell lines), and reports more frequent mutations only, KEAP1(I) (17 cell lines) and KEAP1(O) (48 cell lines).

(B) Mutation matrix for 3 mutations reported by SuperDendrix to be associated with NFE2L2 dependency shows approximate mutual exclusivity between mutations.

(C) Waterfall plot of FANCG dependency scores colored according to presence of the single mutation BRCA1(I) (15 cell lines) reported by SuperDendrix to be associated with FANCG dependency. Most of the cell lines with the largest FANCG dependency have BRCA1(I) mutations. (D) Waterfall plot of the single mutation BRAF(A) (65 cell lines) reported by the univariate test to be associated with FANCG dependency shows that cell lines with largest FANCG dependency do not have BRAF(A) mutations.

The second bunch of results are indeed richer and more interesting however, again the vast majority of hits are represented by individual cancer lineages and transcription factors dependencies, which would probably be unveiled via a simple differential dependency analysis contrasting cell lines from a given tissue to the rest of the panel.

First, we note that the goal of the analysis in the manuscript was not merely to identify cancer-type specific dependencies, but rather to assess whether dependencies are associated with genetic aberrations, cancer types, or some combination of both features. Thus, our analysis examined associations between differential dependencies and **combinations of cancer types and mutations**.

To address the comment that a simple differential dependency analysis would also find similar associations, we conducted a systematic univariate analysis using both types of features - cancer type and mutations - and compared the results to those found by SuperDendrix. We found that the results were analogous to the comparison above of the univariate test and SuperDendrix: the univariate test reported more associations with features (mutations or cancer types) that were more frequently represented in the data and also reported more associations in cell lines with more mutations. At the same time, the univariate test missed associations to rare features such as Eye cancer and BCL2(A) mutation that were reported by SuperDendrix. We describe this analysis in a new section "*Univariate analysis of cancer-type-specific differential dependencies*" of the revised manuscript, which we reproduce below.

"We conducted a systematic univariate analysis to search for associations between differential dependencies and combinations of cancer type and/or mutation features. We analyzed a total of 474,208 pairs consisting of one of 511 differential dependencies and one of 928 features (31 cancer types and 897 mutations). This univariate analysis identified 861 significant associations ($FDR \leq 0.2$) between 334 differential dependencies, 25 cancer types and 142 mutations (Figure S11), compared to 501 significant associations ($FDR \leq 0.2$) between 227 differential dependencies, 27 cancer types and 55 mutations identified by SuperDendrix (Figure S11).

We find a sizable difference between the associations identified by the univariate test and SuperDendrix. While 203 differential dependencies are reported by both methods to have associations (Figure S11A), the univariate test reports an additional 131 unique differential dependencies with associations, while SuperDendrix reports an additional 24 unique differential dependencies with associations (Figure S11B-C). We found that the associations reported uniquely by the univariate test are biased towards frequent features and cell lines with higher mutation rate, analogous to the results reported above with mutation features alone. Specifically, the features in associations reported uniquely by the univariate analysis have a higher average frequency than those in associations reported uniquely by SuperDendrix (univariate: 39.2,

SuperDendrix: 26.7, P -value = 0.002; t -test, Figure S11D). On the other hand, the features in associations reported uniquely by SuperDendrix that were not in associations reported by the univariate test are all rare features that occur in less than 20 cell lines (average frequency: 13.1, starred in Figure S11B). In addition, the difference in the number of associations reported by the univariate test and SuperDendrix is positively correlated ($R = 0.5$, P -value $< 2.2e-16$; Pearson correlation, Figure S11E) with the total number of mutations in the cell line, indicating that some of the associations reported by the univariate test are likely false positives in cell lines with high numbers of mutations. These suggest two issues of the univariate test: The univariate test lacks sensitivity in features with low frequency and specificity in cell lines with many mutations because the univariate test evaluates each feature independently and lacks a procedure to control for variable mutation rate of cell lines. In contrast, SuperDendrix evaluates combinations of mutually exclusive features and controls for mutation rate of cell lines in the statistical test of its third module.

Next, we compared the features that were reported to be associated with the 203 differential dependencies identified by both methods (Figure S11A). We found that both methods reported the same sets of features for 52 of these 203 differential dependencies. Associations reported uniquely by the univariate test tended to include frequent features and cell lines with high mutation rates. Furthermore, the univariate test reported many differential dependencies to be associated with both a mutation and a cancer type where this mutation frequently occurred. For example, BRAF(A) is the mutation with most associations reported by the univariate test, and this occurs frequently in skin cancer (39/65 cell lines with BRAF(A) are skin cancer, fold-enrichment = 8.52, P -value = $1.2e-39$; hypergeometric test, Figure S12A). Interestingly, 24 of the 34 differential dependencies reported by the univariate test to be associated with *either* BRAF(A) or skin cancer are reported as associated with *both* BRAF(A) and skin cancer (Figure S12B). On the other hand, none of the 26 differential dependencies reported by SuperDendrix to be associated with BRAF(A) or skin cancer are associated with both features. This again demonstrates the key difference between the univariate test and SuperDendrix that was described above: the univariate test evaluates each association independently and does not account for correlation between features while SuperDendrix examines mutual exclusivity of features and thus avoids redundant associations of correlated features. This difference is also apparent in the mutation with second most associations, KRAS(A). Cell lines with KRAS(A) mutation are significantly enriched for pancreatic cancer, colon cancer, lung cancer, and bile duct cancer (Figure S12C). 14 of the 34 differential dependencies reported by the univariate test to be associated with KRAS(A) or these four enriched cancer types are associated with both KRAS(A) mutation and at least one of the enriched cancer types (Figure S12D). In contrast, SuperDendrix does not report any redundant associations in 40 differential dependencies associated with KRAS(A) or the enriched cancer types.

Taken together, these results indicate a similar tradeoff in the identification of associations described previously in the comparison of associations to mutations: While the univariate test reports a higher number of associations than SuperDendrix, its associations tend to include redundant associations between correlated features and are also biased towards cell lines with

higher mutation rate. On the other hand, SuperDendrix prioritizes mutually exclusive features and selects the strongest associations, thus reporting fewer and less redundant associations.”

Figure S11 from the revised manuscript: Associations between differential dependencies (rows) and cancer types (columns labeled in gold) and/or mutations (columns labeled in black) identified by SuperDendrix and the univariate test. (A) Associations for differential dependencies reported by both methods. (B) Associations reported uniquely by SuperDendrix. (C) Associations reported uniquely by the univariate test. (D-E) Associations reported uniquely by the univariate test are biased towards frequent features and cell lines with high number of mutations. (D) Cancer types and mutations in associations identified uniquely by the univariate test have higher frequency than those identified uniquely by SuperDendrix (univariate: 39.2, SuperDendrix: 26.7, P-value = 0.002; t-test). Mutation TP53(I) with outlier frequency (495 cell lines) was excluded from comparison. (E) The differences between the number of associations reported by the univariate test and SuperDendrix for each cell line is positively correlated with the number of mutations in the cell line (R=0.5, P-value < 2.2e-16; Pearson correlation).

Figure S12 from the revised manuscript: The univariate test finds redundant

associations between mutations and the cancer types that are enriched for these mutations. (A) 65 cell lines with BRAF(A) mutation are significantly enriched for Skin cancer (39/65 cell lines with BRAF(A), fold-enrichment = 8.52, P-value = $1.2e-39$; hypergeometric test). (B) 24 of the 34 differential dependencies reported by the univariate test to be associated with BRAF(A) or Skin cancer are associated with both the mutation and Skin cancer. In contrast, 0 of the 26 differential dependencies reported by SuperDendrix to be associated with BRAF(A) or Skin cancer is associated with both features. (C) 130 cell lines with KRAS(A) mutation are significantly enriched for Pancreatic cancer (31/130 cell lines with KRAS(A), fold enrichment = 5.4, P-value = $2.7e-24$), Colon cancer (17/130 cell lines with KRAS(A), fold enrichment = 2.8, P-value = $2.7e-6$), Lung cancer (31/130 cell lines with KRAS(A), fold enrichment = 1.7, P-value = 0.0002), and Bile duct cancer (10/130 cell lines with KRAS(A), fold enrichment = 2.3, P-value = 0.002). (D) 14 of the 34 differential dependencies reported by the univariate test to be associated with KRAS(A) or its enriched cancer types are associated with both the mutation and at least one of the enriched cancer types. In contrast, 0 of the 40 differential dependencies reported by SuperDendrix to be associated with KRAS(A) or its enriched cancer types are associated with both the mutation and at least one of the enriched cancer types.

In addition, in several other works many other computational approaches have been proposed to associate combination of genomic events (potentially ME events) to continuous read-outs, particularly drug responses across panel of cell lines.

In conclusion, without a proper follow up experimental validation or an in-silico validation on an independent dataset (as suggested below), and a more rigorous comparison with other existing methods, the current content of this submission is not sounding enough to warrant publication on Cell Genomics and it might be more appropriate for a more specific bioinformatics journal.

The author should have more convincingly shown advantages of SuperDenrix over other much simpler analytical methods.

We addressed these points by performing several additional analyses that demonstrate the advantages of SuperDendrix including:

- (1) SuperDendrix analysis of a newer version (20Q2) of the DepMap data identifies significant associations for 127 differential dependencies that include 36 sets of multiple features;
- (2) SuperDendrix outperforms NormLRT in the identification of previously annotated core essential and priority target genes (SuperDendrix: AUPRC = 0.1, NormLRT: AUPRC = 0.03;).
- (3) SuperDendrix is robust against systematic biases from mutation rate and identifies associations that are not trivially found by simple univariate analyses.
- (4) SuperDendrix identifies associations that are not found by SELECT, a method for analyzing pairs of mutations based on mutual exclusivity.
- (5) 60% of the associations identified by SuperDendrix are validated on an independent dataset of CRISPR screens from the Sanger Institute.

Further details of these updated results are described in response to the specific points of the reviewer below.

Other points:

* The authors should also compare the outcomes of their strategy to identify differential essential genes with the method based on the normLTR score defined in PMID: 28753431

We applied the method based on normLRT score from PMID: 28753431 to the DepMap data and compared it to SuperDendrix. We found that SuperDendrix identifies differential dependencies with higher precision and recall for a reference gene set from PMID: 30971826. We describe this analysis in the "Comparison with NormLRT" section of the revised manuscript, which we reproduce partially below.

"...

SuperDendrix identified 511 2C differential dependencies while NormLRT identified 949 differential dependencies using the same LRT score threshold of 125 from PMID: 28753431. We compare the two lists of differential dependencies to reference gene sets of Sanger priority target genes (PMID: 30971826) and nonessential genes (PMID: 33407829) that were identified based on gene dependency from independent CRISPR screens.

SuperDendrix outperforms NormLRT in identifying known dependencies, achieving both higher precision and recall (Figure S11) for Sanger priority targets (SuperDendrix: 0.1, NormLRT: 0.03; area under the precision-recall curve (AUPRC)). In addition, differential dependencies from SuperDendrix contain fewer nonessential genes than NormLRT differential dependencies (NormLRT: 3.2% (30/949), 2C: 0.8% (4/511)). We consider nonessential genes which are rarely expressed as the negative control gene set since the differential dependencies are unlikely to be non-essential (unexpressed) for cellular activity."

* While mentioning approaches that attempt to identify associations between differential cancer-dependencies/drug-responses and combination of genomic events, thus sets of multiple biomarkers, the approach presented in PMID: 32437684 could be cited and briefly discussed. In addition, the authors should note that many other previously published works described random-forests and penalized regression models associating groups of genomics features to differential cancer dependencies detected using drug response as readouts. The author should mention this, potentially citing some of these works: PMID: 22460905, PMID: 27397505, PMID: 23180760, PMID: 26482930, PMID: 23993102.

We cited these references and added a brief summary of their main points to the Introduction which we quote below.

"...

Several methods have also been developed to identify associations between genomic alterations and cancer dependencies measured from drug response experiments. For example,

Knijnenburg et al.(PMID: 27876821) introduced LOBICO, an integer linear program to identify logical formulas on a small subset of alteration events that explains a drug response. CELLector (PMID: 32437684) uses a binary tree structure to identify ``CELLector signatures`` - sets of most frequent genomic alterations - and applied ANOVA to test correlation of the signatures with drug response. In addition, studies have used elastic net, a penalized linear regression (PMID: 23180760,PMID: 22460905,PMID: 22460902,PMID: 23993102,PMID: 27397505) and random forest (PMID: 27397505) to test correlation between drug response and combinations of alterations.
...”

* How does Superdendrix perform when applied to drug response data instead of gene essentiality scores? an interesting additional study could assess the extent of agreement between dendrix outcomes across gene-essentiality/drug-response, which might serve as the basis to elucidate drug MoA (as explored in PMID: 3262796)

This is an interesting suggestion, but application to drug response data a significant expansion of the scope of the manuscript and thus we left this as future work as we note in the Discussion:

“Finally, SuperDendrix is a general algorithm that can be used to find associations between binary features (e.g. germline or somatic mutations, cell types) and quantitative phenotypes (e.g. drug response, cell size). It would be interesting to analyze these other phenotypes using SuperDendrix, particularly drug response data from The Genomics of Drug Sensitivity in Cancer (GDSC) database (PMID: 23180760), and compare against other methods (PMID: 23338612, PMID: 27876821) that have been designed specifically to identify associations between drug response and genomic features.”

* The author should compare their algorithm and strategy with those presented in PMID: 28756993 or discuss commonalities/differences

The SELECT method (PMID: 28756993) has three major differences from SuperDendrix. First, SELECT examines only correlations between mutations and does not compute associations between mutations and quantitative phenotypes. In contrast, SuperDendrix scores sets of mutations according to their association with a phenotype of interest. Second, SELECT scores pairs of mutations while SuperDendrix evaluates larger sets of mutations. (Note that the SELECT manuscript describes a heuristic to combine pairs into larger sets, but no software is provided to perform this step and the authors of SELECT did not respond to our request for such software.) Finally, SELECT combines all non-synonymous mutations in a gene into a single feature while SuperDendrix separates mutations in a gene into three features: “*activating*”, “*inactivating*”, and “*other*” according to OncoKB annotations.

Despite these differences, we performed an analysis of the 20Q2 DepMap data using SELECT using the following two-step procedure. First, we ran SELECT to identify pairs of mutually exclusive mutations, and then we computed the associations between pairs of mutations

identified by SELECT and differential dependencies identified by SuperDendrix. We ran SELECT using the mutation features derived by SuperDendrix; i.e. “*activating*”, “*inactivating*”, and “*other*” according to OncoKB annotations. We found that the associations found using mutation pairs from SELECT are dominated by four frequently mutated genes in the dataset including the two most frequent mutations (TP53(L) and KRAS(A)) and two other mutations, BRAF(A) and NRAS(A), that are the 34th and 80th most frequent mutations, respectively, among 897 total mutations. Moreover, SELECT missed several well-known associations identified by SuperDendrix including associations between HRAS dependency and mutations in HRAS and between PIK3CA dependency and mutations in PIK3CA. We describe this analysis

in the “*Comparison with SELECT*” section of the revised manuscript, which we reproduce below.

“The SELECT method (PMID: 28756993) has three major differences from SuperDendrix. First, SELECT examines only correlations between mutations and does not compute associations between mutations and quantitative phenotypes. In contrast, SuperDendrix scores sets of mutations according to their association with a phenotype of interest. Second, SELECT scores pairs of mutations while SuperDendrix evaluates larger sets of mutations. Finally, SELECT combines all non-synonymous mutations in a gene into a single feature while SuperDendrix separates mutations in a gene into three features: “*activating*”, “*inactivating*”, and “*other*” according to OncoKB annotations.

Despite these differences, we used SELECT in a two-step procedure to identify associations between mutations and differential dependencies by first running SELECT on the mutation features derived by SuperDendrix and then applying the univariate test to identify associations between differential dependencies and the mutually exclusive mutations reported by SELECT. SELECT identified only 24 pairs of mutations (in a total of 33 genes) that are associated (via Wilcoxon rank-sum test) with 280 differential dependencies, compared to the 87 sets of mutations in 84 genes that are associated with 127 differential dependencies identified by SuperDendrix. Most of the SELECT associations are dominated by mutations in a small number of well-known cancer genes (Figure S14). For example, 46% of the differential dependencies reported by SELECT to have associated mutations are associated with frequent mutations: KRAS(A) (130 cell lines), BRAF(A) (65 cell lines), TP53(L) (495 mutations), or NRAS(A) (48 cell lines). In comparison, these four mutations are associated with only 27% of the differential dependencies reported by SuperDendrix. Overall, we found that the associations reported by SELECT are biased towards frequent mutations and cell lines with higher mutation rate.

Specifically, the mutations in associations reported by SELECT have a higher average frequency than those in associations reported by SuperDendrix (SELECT: 64.8, SuperDendrix: 38.6, P-value = 1.2×10^{-14} ; t-test, Figure S15A). In addition, the difference in the number of associations reported by SELECT and SuperDendrix is positively correlated ($R=0.27$, P-value = 2.02×10^{-14} ; Pearson correlation, Figure S15B) with the total number of mutations in the cell line, indicating that some of the associations reported by SELECT are likely false positives in cell lines with high numbers of mutations. Lastly, SELECT does not find associations to single mutations or sets of three mutations as it only analyzes pairs of mutations. As a result, the majority (54/74) of the associations reported by SuperDendrix that include only a single mutation are not reported by SELECT. These include associations between HRAS dependency

and HRAS mutation and between PIK3CA dependency and PIK3CA mutation which have been reported previously as oncogene additions.”

Figure S14 from the revised manuscript: Associations found using pairs of mutations identified by SELECT are dominated by mutations in a small number of frequently mutated genes. SELECT associations include 24 pairs containing a total of 33 mutations, while SuperDendrix identifies associations between 87 sets containing a total of 84 mutations. Associations for 46% of the differential dependencies reported by SELECT to have associated mutations include four frequent mutations, KRAS(A) (130 cell lines), BRAF(A) (65 cell lines), TP53(I) (495 cell lines), or NRAS(A) (48 cell lines), compared to 27% of the differential dependencies reported by SuperDendrix. The percentages indicate the proportion of differential dependencies that are reported to have associated mutations.

Figure S15 from the revised manuscript: Associations reported by SELECT are biased towards frequent features and cell lines with high number of mutations. (A) Mutations in associations identified by SELECT have higher frequency than those identified by SuperDendrix (SELECT: 64.8, SuperDendrix: 38.6, P-value = 1.2×10^{-14} ; t-test). Mutation TP53(I) with outlier frequency (495 cell lines) was excluded from comparison. (B) The differences between the number of associations reported by SELECT and SuperDendrix for each cell line is positively correlated with the number of mutations in the cell line ($R = 0.27$, P-value = 2.02×10^{-14} ; Pearson correlation).

* The authors claim that they have implemented an interactive tool for visualising/exploring their results, however the link to superdendrix-explorer is broken and I haven't been able to assess this.

We confirmed that the link to the web browser from the manuscript (link: <https://superdendrix-explorer.lrgq.io/>) works on Chrome, Edge, and Safari. We updated the browser with new data and results from the 20Q2 DepMap data.

* The Broad DepMap dataset used in this manuscript is quite outdated. A 20Q2 version is now available. The authors might consider revamping their analysis with this more recent version. Even better, the authors might use the Sanger dependency map CRISPR-screens data

(available at <https://score.depmap.sanger.ac.uk/>) to independently validate the superdendrix associations. Following this, they could reperform their analysis on a recent joint dataset of cancer dependencies integrating both Sanger and Broad CRISPR screens (available at <https://depmap.org/broad-sanger/>) encompassing data for > 700 cell lines, for which precomputed/batch-corrected CERES scores are available.

We thank the reviewer for these suggestions. We applied SuperDendrix to the 20Q2 version of the Broad DepMap (Avena) dataset. In the new analysis, SuperDendrix identified higher numbers of differential dependencies and significant associations. We have edited the manuscript to reflect this update. Following is a brief summary of the new results.

Differential dependencies:

20Q2: SuperDendrix identified 511 differential dependencies. 19Q1: SuperDendrix identified 492 differential dependencies. Overlap: 396 differential dependencies.

Mutation associations:

20Q2: SuperDendrix identified 127 differential dependencies with significant associations (FDR \leq 0.2) to sets of mutations (Figure 2a). 36 of these are associated with sets of multiple mutations. 19Q1: SuperDendrix identified 32 differential dependencies with significant associations. 9 of these are associated with sets of multiple mutations. Overlap: 30 differential dependencies are reported in both to be associated with mutations. 24 of these are associated with the same mutation sets.

The main associations in the manuscript including the KEAP1-NFE2L2 pathway, the RB1 pathway, and the MAPK pathway are retained in the new results.

Cancer type associations:

20Q2: SuperDendrix identified 227 differential dependencies that are significantly associated with sets of cancer types or sets of cancer types and mutations. 140 of these are associated with sets of multiple features. 43 of these are dependencies on transcription factors. 19Q1: SuperDendrix identified 117 differential dependencies with significant associations. 71 of these are associated with sets of multiple features. 41 are dependencies on transcription factors. Overlap: 90 differential dependencies are reported in both. 34 dependencies on the same transcription factors. Associations to 43 differential dependencies are with the same sets of features.

Other main results on cancer type dependencies from 19Q1 including dependencies on core regulatory circuitry genes in neuroblastoma and dependencies in the TCF3 pathway and IGF1R pathway are retained in the new results.

Figure 2A from the revised manuscript: SuperDendrix identifies associations between mutations and 2C differential dependencies in multiple biological pathways. SuperDendrix weights and P -values for 127 2C differential dependencies with significant ($FDR \leq 0.2$) associations with mutations. 36 of these are associated with sets of multiple mutations; e.g. the set $\{KEAP1(O), KEAP1(I), NFE2L2(A)\}$ are mutations that are (approximately) mutually exclusive and associated with increased dependency on NFE2L2

We had previously validated the associations identified by SuperDendrix from the 19Q1 Avana data using CRISPR screen data (Score dataset) from the Sanger Cancer Dependency Map. In the revision, we repeated the validation for new results from the 20Q2 Avana data using the Score dataset as well as the joint dataset that integrates results of the Broad and the Sanger CRISPR screens. In short, many of the associations identified by SuperDendrix in the Avana dataset were validated as significant associations in the Score dataset and the Integrated DepMap dataset. We describe this analysis in the “*Validation on the Sanger CRISPR-Cas9 screen data*” section of the revised manuscript, which we reproduce partially below.

“...

For each association identified by SuperDendrix in the Avana dataset, we compared the dependency scores of cell lines containing at least one of the features with dependency scores of the cell lines without any feature. We excluded the associations for which dependency or

feature data is not available in the Score dataset. We found that associations between 45/110 differential dependencies and mutations and associations between 146/210 differential dependencies and cancer types and/or mutations identified by SuperDendrix are statistically significant in the Score dataset ($P \leq 0.05$; Wilcoxon rank sum test).

We find that many of the associations identified by SuperDendrix that did not validate in the Score dataset are in cancer types that were poorly represented in the Score dataset (Table S5)."

minor points:

- * Figure legends should be self-explicative, it is not clear without reading the main text what the 2C in figure 1B legend refers to
- * I would change the color scheme associated with activating/inactivating mutations in fig1a. This is misleading as it seems to match the background/outlier distributions' distinction.

We have updated the figure legends and color scheme for Figure 1 accordingly.

Referees' report, second round of review

Reviewers' Comments:

Reviewer #1: The authors have addressed my comments

Reviewer #2: The original paper has been strengthened by the authors thoughtful, deliberate parsing a variety of suggestions by the reviewers. They expand their approach using an expanded DepMap dataset, which expands the number of new findings considerably.

They also complete what is most likely the most diligent, step-by-step evaluation of the new approach compared to existing approaches, alternative datasets, and potential confounders, that I have ever seen! This is buried in the STAR methods. In fact the only remaining concern I have is that some of the most interesting work and evaluation may now be buried in the methods as opposed to in the main paper! Examples including comparison with univariate methods, evaluation of the WRN observation, comparison with SELECT and the Sanger dataset etc.

I wonder whether there is a way to bring some of this very careful analysis into the main text. The field really needs this kind of primer.

Great job overall.

Reviewer #3: In response to my previous criticisms about the lack of sufficiently novel findings the authors have extended their analysis to a newer release of the DepMap data, increasing the number of reported differential dependencies that are associated with combinations of mutually exclusive point mutations are not trivially explicable by looking at the status of a single oncogene.

They have also compared the output of their method with that of univariate analysis mining for associations between individual molecular features and differential dependencies reporting a lower number of significant hits but an increased robustness of the SuperDenrix outcomes in terms of ability to collapse into a unique significant module multiple genomic events founds as hits in the univariate analysis.

In addition, the authors have performed a comparison of their method with a systematic univariate analysis where the factors are combinations of cancer types and individual point mutations reaching similar conclusions and enriching the collection of superDenrix significant modules with combinations of genomic events and cancer types.

Finally, my other points have also been satisfactorily addressed by the authors, who also have validated most of their findings on an independent cancer dependency dataset as I suggested.

I do believe that the following final points should be addressed:

- While moving to the 20Q2 release of the depmap dataset the authors find 396 overlapping differential dependencies (over 511 in total on the 20Q2 and 492 on the 19Q1). One would expect that being 20Q2 a superset of 19Q1 there shouldn't be differential dependencies that are private to 19Q1. The authors should discuss this and possibly motivate why about 100 of the 19Q1 associations are missing from the 20Q2 analysis, as this might pose doubts on the robustness and the stability of their method.

- The authors make a strong case around a module that explains WRN dependency consistently with the previous association between WRN and MSI cell line status. Is the differential dependency associated with their module more significant than that associated with MSI? Is there a risk that the status of the identified module is collectively redundant with respect to MSI status of the cell lines. I do believe that this deserves a further exploration and an appropriate discussion. In addition, would it be worthy to include MSI (and possibly other factors, such as cell line doubling time, growth media, etc.) as a variable in the superdendrix analyses?

Authors' response to the second round of review

We are sincerely grateful for all of the reviewers' efforts throughout the revision process as well as their detailed feedback that have helped make significant improvements in our manuscript. We provide a point-by-point response to the additional comments from Reviewer #3 below, and have revised our manuscript to address these comments. Reviewer comments are listed in black text with our responses in blue text.

Reviewers' Comments:

Reviewer #1: The authors have addressed my comments

We thank the reviewer for the effort in reviewing this manuscript.

Reviewer #2: The original paper has been strengthened by the authors thoughtful, deliberate parsing a variety of suggestions by the reviewers. They expand their approach using an expanded DepMap dataset, which expands the number of new findings considerably.

They also complete what is most likely the most diligent, step-by-step evaluation of the new approach compared to existing approaches, alternative datasets, and potential confounders, that I have ever seen! This is buried in the STAR methods. In fact the only remaining concern I have is that some of the most interesting work and evaluation may now be buried in the methods as opposed to in the main paper! Examples including comparison with univariate methods, evaluation of the WRN observation, comparison with SELECT and the Sanger dataset etc.

I wonder whether there is a way to bring some of this very careful analysis into the main text. The field really needs this kind of primer.

Great job overall.

We thank the reviewer for the effort in reviewing this manuscript. We sincerely appreciate the reviewer's positive recognition of our analysis and its contribution. Unfortunately, due to space limitations we are unable to move further details of this analysis into the main text.

Reviewer #3: In response to my previous criticisms about the lack of sufficiently novel findings the authors have extended their analysis to a newer release of the DepMap data, increasing the number of reported differential dependencies that are associated with combinations of mutually exclusive point mutations are not trivially explicable by looking at the status of a single oncogene.

They have also compared the output of their method with that of univariate analysis mining for associations between individual molecular features and differential dependencies reporting a lower number of significant hits but an increased robustness of the SuperDenrix outcomes in terms of ability to collapse into a unique significant module multiple genomic events found as hits in the univariate analysis.

In addition, the authors have performed a comparison of their method with a systematic univariate analysis where the factors are combinations of cancer types and individual point mutations reaching similar conclusions and enriching the collection of superDenrix significant modules with combinations of genomic events and cancer types.

Finally, my other points have also been satisfactorily addressed by the authors, who also have validated most of their findings on an independent cancer dependency dataset as I suggested.

We thank the reviewer for the effort in reviewing this manuscript. We are particularly grateful for the constructive criticisms and detailed suggestions that have strengthened our paper.

I do believe that the following final points should be addressed:

- While moving to the 20Q2 release of the depmap dataset the authors find 396 overlapping differential dependencies (over 511 in total on the 20Q2 and 492 on the 19Q1). One would expect that being 20Q2 a superset of 19Q1 there shouldn't be differential dependencies that are private to 19Q1. The authors should discuss this and possibly motivate why about 100 of the 19Q1 associations are missing from the 20Q2 analysis, as this might pose doubts on the robustness and the stability of their method.

We compared the 20Q2 release of the DepMap dataset with the 19Q1 release to investigate why 96 differential dependencies were identified only in the 19Q1 release.

First, we note that the cell lines in the 20Q2 release are **not** a strict superset of the cell lines in the 19Q1 release. Specifically, while the 20Q2 release contains more cell lines (769 cell lines in 20Q2 vs. 558 in 19Q1), 3.6% of the cell lines (20/558) in the 19Q1 release are **not** in the 20Q2 release. This is because the Project DepMap has dropped or merged some cell lines due to various internal reasons as the datasets are updated. Since SuperDendrix uses the dependency scores of all cell lines to identify differential dependencies, the loss of some cell lines in the 20Q2 release explains why the differential dependencies in the new release are not a superset of those from the previous release.

- The authors make a strong case around a module that explains WRN dependency consistently with the previous association between WRN and MSI cell line status. Is the differential dependency associated with their module more significant than that associated with MSI? Is there a risk that the status of the identified module is collectively redundant with respect to MSI status of the cell lines. I do believe that this deserves a further exploration and an appropriate discussion. In addition, would it be worthy to include MSI (and possibly other factors, such as cell line doubling time, growth media, etc.) as a variable in the superdendrix analyses?

We conducted the proposed analysis of the WRN dependency using the MSI status (available for 639 of 769 cell lines from the DepMap 20Q2 release) as an additional binary feature in the feature matrix of SuperDendrix. We used the MSI status for each cell line reported in Chan et al., 2019 (PMID: 30971823).

We find that WRN dependency is more significantly associated with KMT2B(I) mutation found by SuperDendrix than MSI status (KMT2B(I): 0.0000, MSI: 0.0611; P-value from SuperDendrix). We also confirmed that the strongest association with WRN dependency identified by SuperDendrix is KMT2B(I) when MSI is included in the feature matrix. Interestingly, while most (20/24) of the cell lines with KMT2B(I) mutation contain MSI, we find that KMT2B(I) is more specific to increased dependency on WRN; a higher fraction of the KMT2B(I) mutated cell lines are dependent on WRN than the MSI cell lines (KMT2B(I): 18/24, MSI: 22/41, Figure below).

It is possible that the higher significance and specificity of the association between WRN dependency and KMT2B(I) than MSI indicates that the methylation status of H3 histone mediated by the KMT2B(I) mutation may represent a specific molecular mechanism in MSI status that confers the synthetic lethal interaction with WRN. Another alternative is that the MSI status of some cell lines is incorrect. Further validation studies will be necessary to distinguish the functional linkages between WRN, KMT2B, and MSI.

We added the MSI evaluation to the section “Univariate analysis of the DepMap data” under STAR Methods. Further, we agree that including additional variables in SuperDendrix analysis is a promising idea and added the following sentence to the Discussion:

“Third, our identification of associations did not account for other covariates, although recent studies have demonstrated that CERES scores can be affected by other covariates and confounding variables such as tumor mutation burden, cell doubling time, cell cycle stage, growth media, culture type, etc (PMID: 23770567, PMID: 29625053, PMID: 32687165, PMID: 33651980).”

Figure S10 from the revised manuscript: Comparison of associations between increased dependency on WRN and KMT2B(I) mutation and MSI status. Waterfall plot of WRN dependency for the mutation KMT2B(I) reported by SuperDendrix MSI status of cell lines shows extensive co-occurrence between the two features (20 of 24 cell lines containing KMT2B(I) mutations have MSI) as well as higher specificity of KMT2B(I) for increased dependency on WRN (Fraction of dependent cell lines: KMT2B(I): 18/24, MSI: 22/41).